



**Variations in the East Asian summer monsoon over the past**
**millennium and their links to the Tropic Pacific and North**
**Atlantic oceans**
Fucai Duan[a,*], Zhenqiu Zhang[b,c,*], Yi Wang[d,e,*], Jianshun Chen[a], Zebo Liao[c], Shitao
Chen[c], Qingfeng Shao[c], Kan Zhao[c]
[a]College of Geography and Environmental Sciences, Zhejiang Normal University,
Jinhua 321004, China
[b]School of Life Sciences, Nanjing Normal University, Nanjing 210023, China
[c]College of Geography Science, Nanjing Normal University, Nanjing 210023, China
[d]Department of Geography and School of Global Studies, University of Sussex,
Falmer, Brighton BN1 9QJ, UK
[e]Department of Earth System Science, Institute for Global Change Studies, Tsinghua
University, Beijing 100084, China
*Corresponding authors:
E-mail addresses: fcduan@foxmail.com (F. Duan), zhangzhenqiu163@163.com (Z.
Zhang), yi.wang@sussex.ac.uk (Y. Wang)
**Abstract:** Variations of East Asian summer monsoon (EASM) during the last
millennium could help enlighten the monsoonal response to future global warming.
Here we present a precisely dated and highly resolved stalagmite $\delta^{18}O$ record from the
Yongxing Cave, central China. Our new record, combined with a previously
published one from the same cave, indicates that the EASM has changed dramatically
in association with the global temperature rising. In particular, our record shows that
the EASM has intensified during the Medieval Climate Anomaly (MCA) and the
Current Warm Period (CWP) but weakened during the Little Ice Age (LIA). We find
that the EASM intensity is similar during the MCA and CWP periods in both northern
and central China, but relatively stronger during the CWP in southern China. This
discrepancy indicates a complicated regional response of the EASM to the
anthropogenic forcing. The intensified and weakened EASM during the MCA and
LIA matches well with the warm and cold phases of Northern Hemisphere surface air
temperature, respectively. This EASM pattern also corresponds well with the rainfall
over the tropical Indo-Pacific warm pool. Surprisingly, our record shows a strong
association with the North Atlantic climate as well. The intensified (weakened)
EASM correlates well with positive (negative) phases of North Atlantic Oscillation. In
addition, our record links well with the strong (weak) Atlantic meridional overturning
circulation during the MCA (LIA) period. All above-mentioned correlations indicate
that the EASM tightly couples with oceanic processes in the tropical Pacific and
North Atlantic oceans during the MCA and LIA.
**Keywords:** Stalagmite; East Asian summer monsoon; Global warming; Last



Millennium; Little Ice Age; Medieval Climate Anomaly

## 1    Introduction

The last millennium was climatically characterized by the Medieval Climate
Anomaly (MCA; 900-1400 AD) and the Little Ice Age (LIA; 1400-1850 AD), and the
Current Warm Period (CWP; 1850AD to present). These three episodes attract broad
attention within the scientific and policy-making communities, because they contain
critical information to distinguish between the natural and anthropogenic climate
variability. Origins of the MCA and LIA are attributed to the radiative forcing
associated with solar activities and volcanic eruptions, yet the CWP is considered as a
result of increasing anthropogenic greenhouse gases. In particular, the CWP is much
warmer than the MCA (Man et al., 2009; Chen et al., 2018). In association with the
global temperature change, East Asian summer monsoon (EASM) precipitation has
changed significantly. Many studies have indicated that monsoonal climate of China
has generally recorded wetter MCA and drier LIA in the north but reverse conditions
in the south (Tan et al., 2009; Chen et al., 2015; Xu et al., 2016; Tan et al., 2018).
However, it is unclear about the variation of EASM during the MCA and LIA over
central China. Moreover, less is known about the relative intensity of EASM between
the CWP and MCA, two recent warm periods. The examination of the relative
monsoon intensity is the key to evaluating the monsoon response to the anthropogenic
warming.
To better understand monsoonal responses to the global warming climatic
condition, it is necessary to appreciate the natural forcing of EASM during the MCA
and LIA periods. The EASM is strongly influenced by the Tropical Pacific and North
Atlantic Oceans. The Pacific Ocean feeds the warm and moisture air directly into the
EASM, and therefore exerts a strong influence. Several studies have indicated that the
mean-state of EASM is affected by alternations of La Nina-like and El Nino-like
conditions in the Tropical Pacific during the last millennium (Cobb et al., 2003; Yan et
al., 2011a; Rustic et al., 2015; Chen et al., 2018). However, these studies did not reach
an agreement on how the Tropical Pacific affects EASM. To precisely understand the
EASM dynamics, we need to know which changes in EASM are linked to which
modes of the Pacific atmosphere-ocean circulation during the MCA and LIA in central
China. The North Atlantic signal can be transmitted to other parts of the world



through the Atlantic meridional overturning circulation (AMOC; Bond et al., 2001).
Marine sedimentary records have suggested that strong (weak) AMOC over the warm
Greenland interstadials (stadials) correlated with intervals of enhanced (reduced)
EASM during the last glaciation (Wang et al., 2001; Jiang et al., 2016). Similarly,
weak EASM episodes occurred in association with ice-rafted events in the North
Atlantic, which is capable of weakening the AMOC during the Holocene (Wang et al.,
2005; Zhao et al., 2016). This covariation implies a persistent influence of the AMOC
on EASM. However, there is no direct evidence to support the link between the
AMOC and EASM during the MCA and LIA intervals.

Here we present a new precisely-dated and highly-resolved stalagmite record

from Yongxing Cave, Central China. This record, together with a recently published
records from the same cave (Zhang et al., 2019), advances our understanding of the
EASM dynamics during the last millennium.
## 2   Materials and methods

Two stalagmites (YX262 and YX275) are used in this study, both are from

Yongxing Cave (31°35′N, 111°14′E; elevation 800 m above msl), central China. The
previously published stalagmites YX275 has reported detailed variability in the
EASM since the LIA (Zhang et al., 2019). The new candle-like stalagmite YX262 is
159 mm long and 55 mm wide. The Yongxing Cave is located between the Chinese
Loess Plateau and the Yangtze River. Average annual rainfall is about 1000 mm at the
site of the cave. Atmospheric temperature is about 14.3 ℃ and relative humidity is
close to 100% inside the cave. The cave site is climatically influenced by East Asian
Monsoon, featured with wet and warm summer, and dry and cold winter.

Stalagmite YX262 was first halved and then polished for the purpose of the

subsequent sampling. For stable isotope analyses, powdered subsamples, weighing
about 50-100 μg, were drilled on the polished surface along the central growth axis of
the stalagmite. A total of 159 subsamples were obtained at 1 mm increments. The
$\delta^{18}O$ measurements were performed on a Finnigan-MAT-253 mass spectrometer at
Nanjing Normal University. Results are reported as per mil (‰) against the standard
Vienna Pee Dee Belemnite (VPDB). Precision of $\delta^{18}O$ is 0.06‰ at the 1-sigma level.
For U-Th dates, six powdered subsamples, about 100 mg each, were drilled along the
central growth layer. Procedures for chemical separation and purification of uranium


and thorium were described in Shao et al. (2017). U and Th isotope measurements
were performed on a Neptune MC-ICP-MS at Nanjing Normal University. All the
dates are in stratigraphic order with uncertainty of less than 0.03% of the actual dates.

# 3  Results

## 3.1  Chronology

The six U-Th dates and corresponding isotopic rations are shown in Table 1.
Adequate uranium concentrations (0.5–0.7 ppm) and low initial thorium contents
(200–700 ppt, with the exception of 1440 ppt) produced precise dates with small age
uncertainty (6–20 years). The chronology for the stalagmite was established by the
StalAge algorithm (Scholz and Hoffmann, 2011). The age model shows that the
stalagmite YX262 was deposited from 1027 to 1639 AD (see Fig. 2). The age-depth
plot indicates the growth rate of the stalagmite is stable, reaching 0.26 mm/year. The
high and stable growth rate suggests that the stalagmite grew continuously without a
significant hiatus. Visual inspections consolidate the continuity of the stalagmite
growth. The temporal resolution is 3.8 year, allowing for detailed characterizing the
Asian hydroclimate for the first half of the second millennium.

## 3.2  Stable isotope

The $\delta^{18}O$ record of YX262 displays a pronounced fluctuation during the whole
period (see Fig. 3). The $\delta^{18}O$ values ranges from -9.31‰ to -7.88‰, averaging
-8.60‰. The $\delta^{18}O$ values decrease gradually from 1027 to 1372 AD, and then increase
gradually before rapidly increasing to the $^{18}O$-enriched conditions from 1515 AD. The
interval with high $\delta^{18}O$ values is ~100-year long, which is terminated by a pulse to
more negative values at 1626 AD. In general, the $^{18}O$-depleted interval is coeval with
the MCA and the $^{18}O$-enriched interval corresponds to the early LIA (see Fig. 3).

# 4  Discussion

## 4.1  The interpretation of our $\delta^{18}O$

Stalagmite YX262 was deposited under the condition of isotope equilibrium.
Relative to the Hendy tests, replication tests have been considered as a more vigorous
method to examine the isotope equilibrium (Dorale and Liu, 2009). The YX262 $\delta^{18}O$
record matches another Yongxing cave record during the overlapping interval (see Fig.





4; Zhang et al., 2019), indicating an equilibrium condition for the isotope. Thus, the
YX262 $\delta^{18}$O signal is less influenced by the kinetic fractionation and is primarily of
climatic origin. Nevertheless, the climatic significance of the cave $\delta^{18}$O record in
eastern China remains a long-term scientific debate. The cave $\delta^{18}$O records are
normally interpreted as large-scale and integrated changes in the Asian summer
monsoon intensity (e.g., Wang et al., 2001; 2005; Cheng et al., 2009; 2016). This
interpretation is supported by strong correlations among the cave $\delta^{18}$O records across
China (e.g., Yuan et al., 2004; Zhao et al., 2010; Li et al., 2014), and by covariations
of the $\delta^{18}$O records with other proxy records reflecting the monsoon intensity or local
rainfall (e.g., Goldsmith et al., 2017; Zhao et al., 2015; Owen et al., 2016). However,
some studies have revealed that the calcite $\delta^{18}$O records reflect changes in moisture
sources (e.g., Pausata et al., 2011), in particular, regarding the lower (higher) $\delta^{18}$O
values derived from Indian Ocean-dominated (Pacific Ocean-dominated) moisture
sources (e.g., Maher and Thompson, 2012; Tan, 2014). Two most recent studies have
reconciled these two contradictory interpretations (Orland et al., 2015; Wang et al.,
2018). They found that the Chinese stalagmite $\delta^{18}$O records documented a
combination of changes in the isotopic fractionation of water vapor sourced from the
Indian and/or Pacific Oceans, and changes in summer monsoon intensity. In reality,
the extent of the isotopic fractionation of water vapor from the tropical oceans reflects
the changes in integrated monsoon rainfall between the tropical oceans and cave sites
(Yuan et al., 2004). Thus, the stalagmite $\delta^{18}$O signal reflects the regional summer
monsoon intensity (Orland et al., 2015; Tan et al., 2015), with lower $\delta^{18}$O values
reflecting stronger monsoon and higher $\delta^{18}$O values weaker monsoon (Cheng et al.,

2016).

**4.2    The regional characters of the MCA and LIA**

The climate condition during the MCA and LIA has been extensively studied for

the monsoonal China (e.g., Chen et al., 2015; Xu et al., 2016; Tan et al., 2018). In
general, wetter in the north and drier in the south were inferred during the MCA and
the opposite during the LIA (Chen et al., 2015; Tan et al., 2018). The boundary
between the north and south of China was estimated to be about along the River Huai
at 34 °N (Chen et al., 2015), the modern geographical dividing line between northern
and southern China. As an interesting exception, the Dongge cave records in Guizhou,
Southwestern China (25°17′N, 108°5′E) showed a wetter MCA and drier LIA (see Fig.



3; Wang et al., 2005; Zhao et al., 2015). This is consistent with strong spatiotemporal
variability of precipitation in the broad EASM region. Here our Yongxing record,
slightly south to 34 ⁠°N, shows a similar condition as the Dongge Cave (see Fig. 3). As
illustrated in Fig. 3, the stalagmite records from the two caves show a general
similarity in shape and thus each of them truthfully registers the broad climate signal.
An extra comparison shows that the Yongxing and Dongge records in the south vary
broadly in agreement with the Wanxiang (Zhang et al., 2008) and Huangye (Tan et al.,
2011) records in the north, indicating the wetter MCA and drier LIA (see Fig. 3).
However, a minor but important discrepancy exists between the northern and southern
cave records during the MCA. The cave records in the south display an increasing
monsoon trend, but those in the north reflect a decreasing monsoon trend during the
MCA (see Fig. 3 for trends indicated by the arrows). To explain this discrepancy, we
compare all our cave records to changes in temperatures of Northern Hemisphere
(Mann et al., 2009) and northern China (Tan et al., 2003), and meridional
displacement of the Intertropical Convergence Zone (ITCZ; Haug et al., 2001). The
result indicates that the cave records in the south and north collectively exhibit a
broad similarity to the variation in the temperatures and the displacement of the ITCZ
(see Fig. 3). Detailed inspection displays that the weakening monsoon signal recorded
in the northern caves of China during the MCA parallels with the decreasing
temperatures in the Northern Hemisphere and northern China. In contrast, the
intensified monsoon signal recorded in the southern caves of China during the MCA
corresponds to the northward displacement of the ITCZ. The comparison indicates
that the different climate patterns between the south and north may result from
different controlling factors at lower and higher latitudes, respectively. It seems that
the cold temperature from the north restrains the northward migration of the
monsoonal rain belt related to the movement of the ITCZ during the MCA, leading to
the hydrological seesaw between the north and south. It is noted that the enhanced
monsoon condition documented in the Yongxing and Dongge records is contradictory
with those reported in many other paleoclimate records in the south. For example,
drier MCA and wetter LIA were suggested in an integrated stalagmite $\delta^{18}O$ record
from Sichuan Province (Tan et al., 2018), a pollen-derived rainfall record near the
Yongxing Cave site (He et al., 2003), and a lake-based rainfall record in Guangdong
Province (Chu et al., 2002). This regional discrepancy can be checked by additional
highly-resolved and precisely dated records in southern China.





### 4.3 The monsoon intensity during the MCA as compared to the CWP

A comparison of the relative intensity of EASM between the MCA and CWP could be useful to evaluate the response of EASM towards the current global warming. Many studies have found that the CWP is much warmer than the MCA on global and hemispheric scales (Bradley et al., 2003; Mann et al., 2008, 2009; PAGES 2k Consortium, 2013). With regard to the hydrological response, northern China shows a stronger or comparable monsoon condition during the MCA as compared to the CWP (e.g., the Wangxiang and Huangye Caves' records in Fig. 3). A similar monsoon condition is also documented in the Yongxing record in central China (see Fig. 4). However, two Dongge records in southern China collectively shows a slightly weaker monsoon condition during the MCA as compared to the CWP (see Figs. 3, 4). This is indicated by an overall 0.39‰ higher $\delta^{18}O$ value during the MCA than the CWP (Fig. 4). The stronger monsoon condition during the CWP relative to the MCA is parallel to the global temperature evolution, in particular in the western Pacific Warm Pool region (Chen et al., 2018). This correspondence supports the hypothesis that current global warming intensifies the Asian summer monsoon (Wang et al., 2013). The intensified Asian summer monsoon was suggested due to strong coupling of the climate system related to the global warming. Wang et al. (2013) have stated a mega ENSO condition could trigger a stronger EASM in the CWP through the intensified Hadley and Walker circulations. On the other hand, southern China is partially influenced by the Indian Ocean, which also brings moisture to the area of our study (An et al., 2011). We suggest the small discrepancy between Yongxing and Dongge records could be due to the different localized effects in southern China as Dongge Cave is much closer to Indian Ocean than Yongxing Cave.

Different scenarios exist in the South China Sea regarding to the hydrologic variation between the MCA and CWP. The South China Sea is climatically influenced by the EASM and tropical Pacific climate. The lacustrine and coralline records collectively indicate a comparative climate condition between the MCA and CWP (Yan et al., 2011b; Deng et al., 2017). The MCA and CWP are considered to be drier than the LIA in the South China Sea. Yan et al. (2011b) highlighted that a decrease and eastward shift of the Pacific Walker circulation were responsible primarily for the drier climate condition during the MCA and CWP. However, changes in the Walker circulation (Yan et al., 2011b) are in contrast to other estimations (Wang et al., 2013; Cobb et al., 2003), which suggested a strong Pacific Walker circulation during the





warm periods. Due to the contradiction on the Pacific Walker circulation changes, the
trigger for the intensified Asian monsoon during the CWP needs further verification.
Therefore, continued studies are needed on the links between the EASM and the
Pacific climate.
**4.4    The link to the Tropical Pacific Ocean**

The ITCZ and El Niño-Southern Oscillation (ENSO) exert profound influences

on the precipitation in East Asia during the last millennium (Wang et al., 2013). As
shown in Fig. 5, our calcite record shows a great similarity to temperature and
hydrology reconstructions over the tropical Indo-Pacific warm pool (IPWP).
High-resolution sediment (Oppo et al., 2009) and speleothem (Griffiths et al., 2016)
records over the IPWP collectively suggest warm sea surface temperatures and
reduced rainfall during the MCA and CWP, and reversed conditions during the LIA
(Fig. 5). The rainfall over the IPWP is anti-phased with the EASM intensity,
supporting the modulation of the ITCZ' latitudinal migration on the EASM during the
last millennium (Zhao et al., 2015; Xu et al., 2016; Griffiths et al., 2016). In addition,
the temperature change over the IPWP can influence the EASM intensity via the
expansion and contraction of the ITCZ (Yan et al., 2015; Chen et al., 2018). The warm
MCA and cold LIA conditions do not necessarily signify a La Nina-like condition
during the MCA and an El Nino-like condition during the LIA over the IPWP.
Conversely, rainfall-based ENSO reconstructions showed the El Nino- and La
Nina-like conditions during the MCA and LIA, respectively (Moy et al., 2002, Yan et
al., 2011a; Fig. 5e, f). The sediment-derived ENSO variation in Ecuador (Moy et al.,
2002) and the composite ENSO reconstruction across the Tropic Pacific (Yan et al.,
2011a) showed a great similarity among the ENSO signals and the timing of switches
between the ENSO cold and warm phases. These ENSO reconstructions resemble
well with Yongxing records (Fig. 5). For example, the El Nino- and La Nina-like
conditions during the MCA and LIA parallel with the intensified and weaken EASM
from the Yongxing Cave, respectively. In particular, the switch of the ENSO phases
from the MCA to LIA coincides with the EASM intensifying peak during the MCA
(Fig. 5). These strong correlations indicate a dynamical link between the EASM
intensity and ENSO modes. In the summer after the El Niño evolves to maturity, an
abnormally blocked anticyclone takes place in Northeast Asia. At the same time, the
subtropical high in the western North Pacific extends westward abnormally. This
abnormal circulation pattern strengthens the EASM in subtropical East Asia (Wang et



al., 2001). Despite the potential monsoon-ENSO link, the ENSO reconstructions still
need further verification due to their different variations. A recent temperature record
in eastern equatorial Pacific (Rustic et al., 2015) supports the rainfall-based ENSO
reconstruction (Moy et al., 2002; Yan et al., 2011a), with the El Nino- and La
Nina-like mode during the MCA and LIA, respectively. This record challenges the
paradigm of the La Nina-like pattern during the MCA followed by the El Nino-like
pattern during the LIA (Cobb et al., 2003). However, the study of Rustic et al. (2015)
showed the strongest El Nino-like situation occurred at the late MCA to early LIA
transition, instead of the peak MCA.

**4.5   The link to the North Atlantic Climate**

Surprisingly, our Yongxing record shows a good correlation with the North
Atlantic climate. As illustrated in Fig. 6, the intensified (weakened) EASM during the
MCA (LIA) coincides with a persistent positive (neutral to slightly negative) North
Atlantic Oscillation index (NAO; Trouet et al., 2009; Fig. 6c). In addition, these
EASM variations resemble changes of the Atlantic meridional overturning circulation
(AMOC), measured by the drift ice index (Bond et al., 2001; Fig. 6d) and mean grain
size of sortable silt (Fig. 6e; Thornalley et al., 2018) in the North Atlantic. The
intensified EASM corresponds to the strong AMOC during the MCA and the
weakened EASM to the weak AMOC during the LIA, which is consistent with the
scenario during the last glaciation (Wang et al., 2001; Böhm et al., 2015). These
strong correlations indicate an influence of the NAO and AMOC on the EASM.
During the MCA, positive NAO induces a warmer winter in Europe, which reduces
snow accumulation over Eurasia and therefore allows for a penetration inland of the
EASM next summer (Overpeck et al., 1996). Robust AMOC can intensify the EASM
through northward positioning the ITCZ (Wang et al., 2017). During the LIA, weaker
NAO and AMOC would produce decreased EASM in the reversed fashion. It has
been proposed that conditions of the NAO were dynamically coupled to states of the
AMOC (Trouet et al., 2009; Wanamaker et al., 2012). The strong (weak) NAO during
the MCA (LIA) contributes to enhanced (weakened) AMOC through enhancing
(weakening) the westerly (Trouet et al., 2009). Solar activity is usually considered as
the root trigger of natural climate change. The Yongxing record is broadly similar to
change in solar irradiance (Steinhilber et al., 2009; Fig. 6a). The intensifying EASM
is paralleled with the greater solar activity during the MCA and the weakened EASM

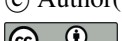



with the less solar activity during the LIA. The solar forcing of the EASM can be
conducted through modulating the Asia-Pacific temperature contrast (Kutzbach et al.,
2008), the AMOC intensity (Wang et al., 2005) and the ENSO condition (Asmerom et
al., 2007; Zhao et al., 2016). However, relative importance of these forcing pathways
is unknown and, most importantly, the ENSO condition remains a matter of debate
during the last millennium (e.g., Cobb et al., 2003; Yan et al., 2011a). As a counterpart
to the MCA, the CWP is similarly marked by intensified EASM, strong AMOC and
high solar output (Fig. 6). However, the relationship between the EASM and NAO
becomes unclear during the CWP, with the intensified EASM failing to match the
expected more positive NAO. Longer term data is needed to assess the linkage
between NAO and EASM during the CWP.

## 5 Conclusions

Based on a new and published stalagmite records from the Yongxing cave,
central China, we reconstruct a continuous evolutional history of the EASM during
the past millennium and link its variation with the Pacific and North Atlantic climates.
The climatic features in our record are generally in agreement with those in the
Wanxiang and Huangye cave records in northern China as well as the Dongge cave
record in southern China. The agreement consolidates our EASM reconstruction by
the Yongxing records. The intensified (weakened) EASM during the MCA (LIA)
correlates with the warm (cold) surface temperature and enhanced (reduced) rainfall
over the IPWP. Based on the strong correlation with the ENSO reconstruction, our
records support an El Nino- like condition during the MCA and a La Nina-like
condition during the LIA. In addition, our records show a potential link between the
EASM and the North Atlantic climate. The intensified EASM coincides with positive
NAO and robust AMOC during the MCA, while the weakened EASM corresponds
with neutral to negative NAO and weak AMOC during the LIA.

## Acknowledgments

This work was supported by Zhejiang Provincial Natural Science Foundation (no.
LY19D020001) and National Natural Science Foundation of China grants (nos.
41602181, 41572340 and 41572151).



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


**Table and figure**

Table l U-series dating results of stalagmite YX262 from Yongxing Cave

| Sample depth (mm) | $^{238}$U (ppb) | $^{232}$Th (ppt) | $\delta^{234}$U (measured) | $^{230}$Th/$^{238}$U (activity) | $^{230}$Th Age (a) (uncorrected) | $\delta^{234}$U$_{initial}$ (corrected) | $^{230}$Th Age (a) (corrected) |
|---|---|---|---|---|---|---|---|
| YX262-5 | 546.0 ±0.5 | 307.9 ±0.6 | 607.5 ±1.0 | 0.006230157 ±0.00014 | 423.5 ±9.4 | 608.2 ±1.0 | 413.1 ±10.8 |
| YX262-25 | 595.5 ±03 | 280.5 ±0.6 | 790.6 ±1.9 | 0.00788248 ±0.00008 | 481.0 ±5.1 | 791.7 ±1.9 | 473.1 ±6.3 |
| YX262-48 | 506.3 ±03 | 281.6 ±0.5 | 762.1 ±1.9 | 0.009468079 ±0.00010 | 587.3 ±6.4 | 763.4 ±1.9 | 577.9 ±8.0 |
| YX262-75 | 517.7 ±03 | 724.3 ±0.1 | 680.5 ±2.1 | 0.010930422 ±0.00010 | 711.3 ±6.4 | 681.8 ±2.1 | 686.3 ±13.9 |
| YX262-95 | 651.8 ±03 | 1448.0 ±0.3 | 806.5 ±2.0 | 0.013146471 ±0.00010 | 796.0 ±6.3 | 808.3 ±2.0 | 759.4 ±19.1 |
| YX262-116 | 583.4 ±08 | 283.0 ±0.4 | 956.6 ±1.0 | 0.014987259 ±0.00012 | 838.0 ±6.6 | 958.9 ±1.0 | 830.8 ±7.5 |

Decay constant values are $\lambda_{234}$=2.82206×10$^{-6}$a$^{-1}$, $\lambda_{238}$=1.55125×10$^{-10}$a$^{-1}$, $\lambda_{230}$=9.1705×10$^{-16}$a$^{-1}$ and $\delta^{234}$U =
([$^{234}$U/$^{238}$U]$_{activity}$-1)×1000. Corrected $^{230}$Th age calculation, indicated in bold, is based on an assumed initial
$^{230}$Th/$^{232}$Th atomic ratio of (4 ±2) × 10$^{-6}$. All corrected dates are years before 2017 A.D.






















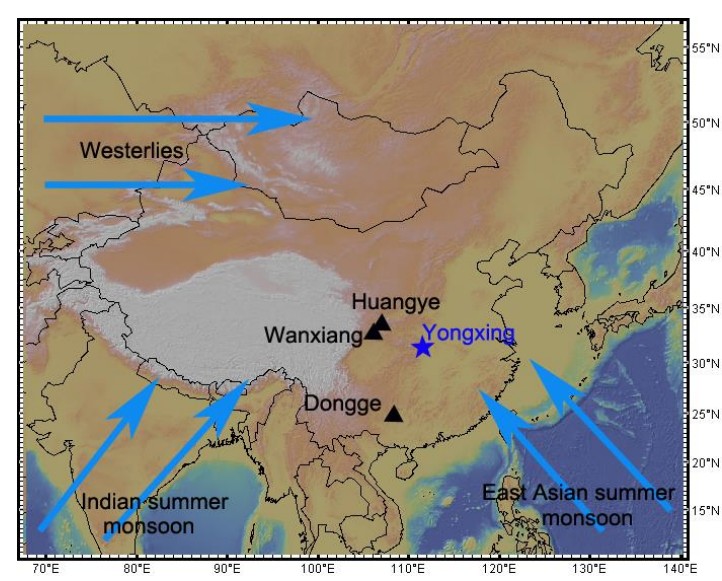

Fig.1 Schematic climate setup of East Asian Monsoon and our study site. The blue
star and black triangles represent Yongxing Cave in central China and other caves in
the monsoonal region, respectively.






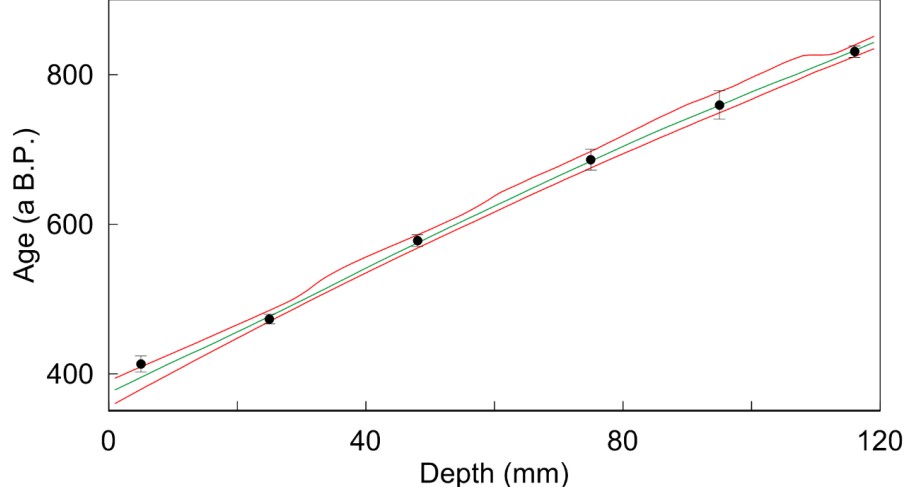


Fig. 2 Age versus depth model for our stalagmite YX262. The black dots and vertical
error bars indicate $^{230}$Th dates and errors of these dates, respectively. The middle
green line indicates the model age, and upper and lower red lines indicate the age in
95% confidence level, respectively.





Fig. 3 A comparison of the Yongxing δ¹⁸O time-series with other proxy records. (a) Northern Hemisphere reconstructed temperature (Mann et al., 2009); (b) Northern China reconstructed temperature (Tan et al., 2003); (c) Huangye Cave δ¹⁸O composite (Tan et al., 2011); (d) Wanxiang Cave δ¹⁸O record (Zhang et al., 2008); (e) Yongxing





Cave record (this study); (f) Dongge Cave record (Wang et al., 2005; Zhao et al.,
2015); (g) Cariaco Basin Ti content record (Haug et al., 2001). Light yellow and blue
bars indicate the MCA and LIA, respectively. Arrows indicate trends of the climatic
variations.

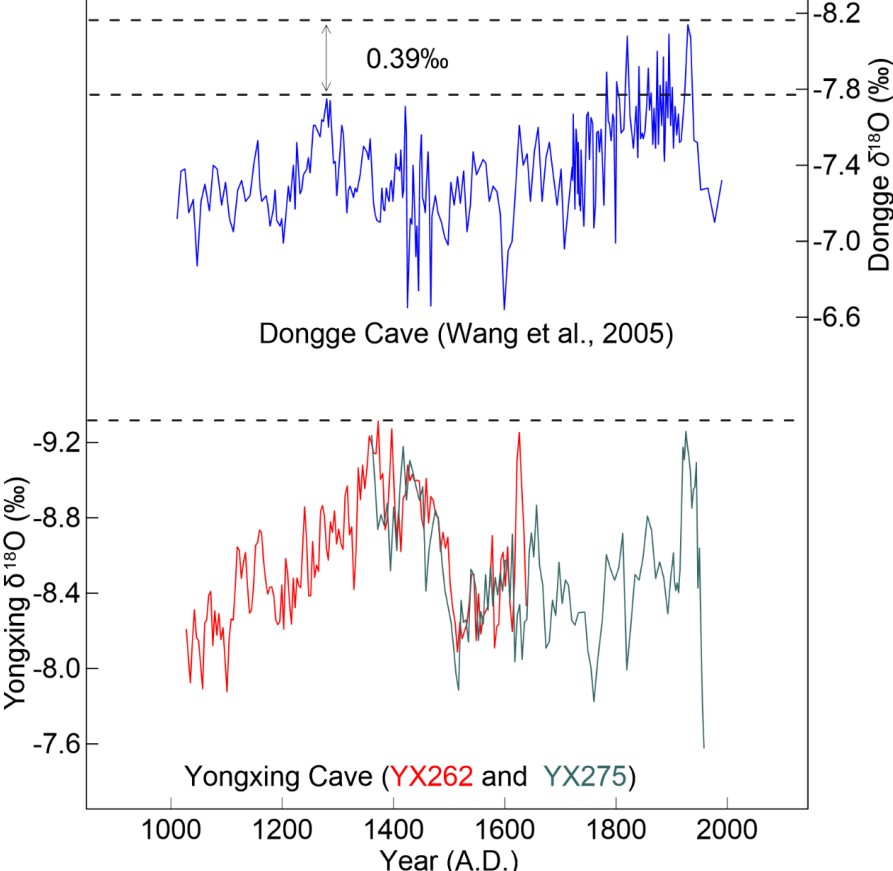


Fig. 4 Relative intensity of EASM during the MCA as compared to the CWP. The
upper panel is the Dongge cave record (Wang et al., 2005); the lower panel is the
Yongxing Cave YX262 (red) and YX275 (green, Zhang et al., 2019) records. On
average, the Dongge Cave record shows a 0.39‰ lower $\delta^{18}O$ values during the CWP
than the MCA. However, the Yongxing record shows a comparable value between the
CWP and MCA.



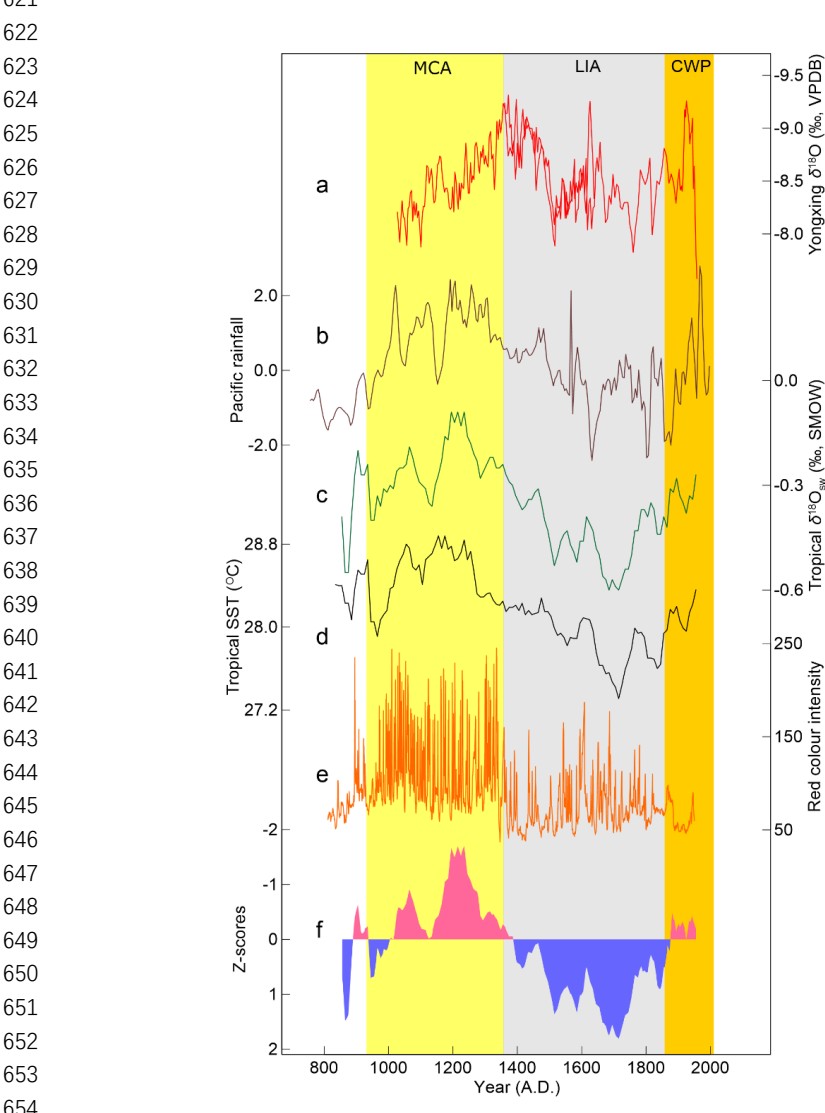

Fig. 5 A comparison between EASM and Pacific climate. (a) Yongxing cave record (this study); (b)Tropical Pacific rainfall record (Oppo et al., 2009); (c) Tropical Pacific δ$^{18}$O record (Oppo et al., 2009); (d) Tropical Pacific sea surface temperature (Oppo et al., 2009); (e) Red colour intensity in southern Ecuador (Moy et al., 2002); (f) Hydrological reconstruction of ENSO from Tropical Pacific (Yan et al., 2011a). Yellow, grey and orange bands represent the MCA, LIA, and CWP, respectively.



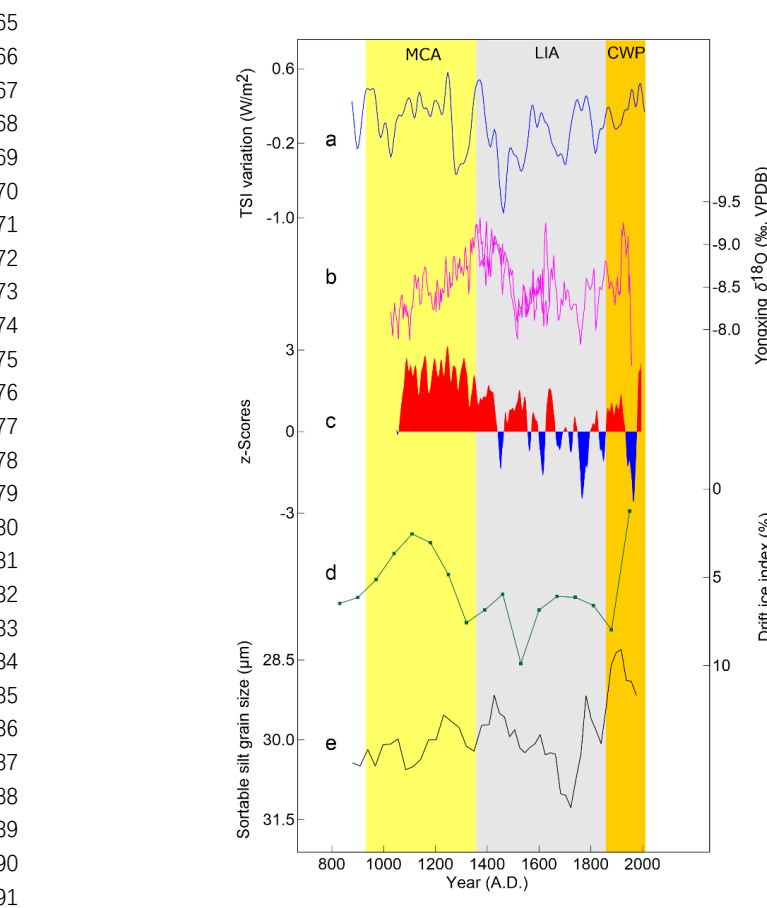

Fig. 6 A comparison between EASM, solar activity and North Atlantic climate. (a) Total solar irradiance (Steinhilber et al., 2009); (b) Yongxing Cave record (this study); (c) North Atlantic Oscillation (Trouet et al., 2009); (d) North Atlantic drift ice index (Bond et al., 2001); (e) Sortable silt grain size in the North Atlantic (Thornalley et al., 2018). Yellow, grey and orange bands represent the MCA, LIA, and CWP, respectively.