# Peer review of "Variations in the East Asian summer monsoon over the past"

_Climate of the Past, 2019_

## Referee Comment (RC1) · Anonymous Referee #1 · 18 Aug 2019

Dear editor and authors of the manuscript "Variations in the East Asian summer monsoon over the past millennium and their links to the Tropic Pacific and North Atlantic oceans", a speleothem with high resolution is an important contribution in the paleoclimate community, if the raw data can be archived in the published repository. Another new information is to discuss the difference between the northern and southern cave records. I am not specialist in speleothem record, and cannot assess the physical meaning of the speleothem oxygen-isotope. However, I am worried about the definition of the East Asian summer monsoon (EASM). As we known, the EASM is highly variable in the meridional direction. A strong reason is necessary to explain that the few records can reflect EASM variation. Moreover, the past millennium includes the 20th century, it is highlighted to quantitatively compare EASM variation between the proxy reconstruction and the instrumental dataset. Thus, I suggest that the manuscript should be accepted for publication after a revision.

Main comments:
1. The speleothem oxygen-isotope has high resolution, thus, its upward or downward trends over some specific time periods need to be quantified using the trend test methods e.g. Mann-Kendall non-parametric trend test. Moreover, the magnitude and amplitude of the EASM intensity also need to be calculated.
2. According to the background of the co-authors, a more mechanism of EASM variation should be discussed. e.g. how the NAO effects the East Asian summer monsoon, based on some instrumental datasets or CMIP5 datasets.
3. The other proxy reconstruction from tree-ring [*Liu et al.*, 2019] and historical documentary [*Ge et al.*, 2008] are suggested to cross check the speleothem EASM reconstruction. Moreover, a detailed and independent local temperature reconstruction should be used to explore the relationship between the speleothem oxygen-isotope and the temperature, e.g. [*Cook et al.*, 2013; *Shi et al.*, 2015; *Zhang et al.*, 2018].

Specific Comments:
1. Page 1, Lines 25-26. What meaning is the 'EASM intensity'? the amplitude or magnitude of the EASM variation?
2. Page 2, Line 49. It is 'Mann'. Moreover, Chen (2018) is not the temperature reconstruction.
3. Page 2, Lines 63-67. The detained review about the disagreement of the influence ENSO on EASM is very interesting and suggested to help the following analysis.
4. Page 3, Lines 78-79. The 'direct evidence' is not rigorous, even it is still difficult to connect the AMOC and EASM for the instrumental dataset analysis.
5. Page 4, Lines 132-133. There is a large discrepancy between YX262 and YX275 in the early 1600s. An discussion of this difference is suggested to indicate an stable condition of the isotope.
6. Page 5, Lines 141-142. The EASM intensity is not equal to the local rainfall, e.g. the increasing meiyu rainfall means the weak EASM.
7. Page 6. Lines 188-189. The statement is inaccurate, since the north-drought and south-flood can be affected by the same factor from the instrumental analysis.

8. Page 7, Lines 214-215. In fact, the EASM becomes weak since the late 1970s [*Wang*, 2001].

9. Pages 7-8. Lines 229-235. When you check Walker cell, the position of ascending or sinking branch is also important for atmosphere transport.

10. Pages 9-10, Lines 278-311. The relationship between NAO and EASM during the CWP is complex, why it is stable during the LIA or MCA. Is a possible reason the uncertainty of NAO reconstruction?

References:

Cook, E. R., P. J. Krusic, K. J. Anchukaitis, B. M. Buckley, T. Nakatsuka, and M. Sano (2013), Tree-ring reconstructed summer temperature anomalies for temperate East Asia since 800 CE, *Clim. Dyn.*, *41*(11-12), 2957-2972.

Ge, Q., X. Guo, J. Zheng, and Z. Hao (2008), Meiyu in the middle and lower reaches of the Yangtze River since 1736, *Chin. Sci. Bull.*, *53*(1), 107-114.

Liu, Y., et al. (2019), Anthropogenic Aerosols Cause Recent Pronounced Weakening of Asian Summer Monsoon Relative to Last Four Centuries, *Geophys. Res. Lett.*, *46*(10), 5469-5479.

Shi, F., et al. (2015), A multi-proxy reconstruction of spatial and temporal variations in Asian summer temperatures over the last millennium *Clim. Change 131*(4), 663-676.

Wang, H. (2001), The weakening of the Asian monsoon circulation after the end of 1970's, *Adv. Atmos. Sci.*, *18*(3), 376-386.

Zhang, H., et al. (2018), East Asian warm season temperature variations over the past two millennia, *Scientific Reports*, *8*(1), 7702.

---

## Referee Comment (RC2) · Anonymous Referee #2 · 8 Sep 2019

The manuscript entitled "Variations in the East Asian summer monsoon over the past millennium and their links to the Tropic Pacific and North Atlantic oceans" by Duan et al. presents a new high-resolution stalagmite d18O record (YX262) form Yongxing Cave, central China over the past millennium. This record, combined with a published record from the same cave (YX275), is used to investigate the relationship between the East Asian summer monsoon (EASM) and Tropic Pacific and North Atlantic oceans during the period. They suggested that the EASM was intensified during the Medieval Climate Anomaly (MCA) and the Current Warm Period (CWP), but wakened during the Little Ice Age (LIA). These observed EASM variations were causally liked to the precipitation change in the tropical Pacific (or ENSO), as well as in the North Atlantic (e.g., NAO

and AMOC). While the d18O record itself appears to be robust, the interpretations are rather tentative and sometimes confusing. I have a few suggestions/comments for improvement pending on which I recommend acceptance of this paper.

General comments:

(1) In the introduction, in order to better introduce research background to readers, it is necessary to add some more references. For example, after the sentence of Lines 46-48, Lines 49-51 and Lines 61-63. In addition, some references are not properly used (not the most proper one).

(2) Please add more information about the stalagmite sample. For instance, the sample image can be added in the figure 2, including subsample locations of U-Th dates and if possible stable isotopes as well. Was the whole sample (YX262) or only one portion analyzed in this study ? What is the mineral of the sample? The authors only mentioned 'calcite record' in the discussion (Line 241).

(3) For discussion 4.1, in Lines 153-156, "Thus, the stalagmite d18O signal reflects the regional summer monsoon intensity......", how to understand the term "regional summer monsoon intensity" ? In addition, the authors should always point out the timescale when they discuss the significance of the stalagmite d18O proxy. In Lines 146-148, "two most recent studies have reconciled these two contradictory interpretations......", it sounds like that the two studies already resolved the debates of the Chinese stalagmite d18O proxy. There are many papers that have addressed to some extent this issue recently, such as Zhao et al., 2018 and Zhang et al., 2019. Additionally, it appears that the cave d18O was considered to be the 'monsoon intensity' and local precipitation as well at different places, lacking a consistency.

(4) The small amplitude changes in stalagmite d18O value may have complicated mechanisms behind, such as temperature effect, amount effect, source changes, upper stream rainout, and evaporations etc. If explained solely as local rainfall amount, please provide a comparison to the instrumental record for each cave record or cite

related published papers.

(5) In the section 4.2, the authors should compare their record with the stalagmite record from Heshang Cave, which is fairly close to Yongxing Cave. I suggest adding the Heshang d18O record in the figure 3, and have a related discussion in the section. In addition, I strongly encourage the authors to compare the Yongxing record with local historical records or cite related papers, which may provide a validation test on the interpretation of the Yongxing d18O record.

(6) When comparing the d18O values between different time periods (the MCA, LIA and CWP), the differences of the mean values should be provided. The trends of records, as well as similarities between different records are merely visually defined. Statistical methods should be considered to show their significances.

(7) In sections 4.4 and 4.5, I suggest that the authors analyze the relationship of the local precipitation (and/or d18O) at Yongxing Cave site with ENSO, NAO, PDO and AMOC indexes (reconstructed from instrumental data).

(8) Overall, the causal links of the stalagmite d18O records with the AMOC, NAO, ENSO as suggested by the authors are rather tentative. For example, a visual similarity between two records cannot be used to definitively validate their causal linkage.

Specific comments:

Lines 24-26: I don't think we can say the "EASM intensity is similar in both northern and central China, ......". I mean we cannot say local EASM intensity instead of local precipitation amount. In addition, the timescale should be always mentioned.

Lines 31 and 278: The authors use "surprisingly" twice in the manuscript. Actually, many studies already found the North Atlantic climate can influence the EASM changes, for example He et al. (2017).

Lines 74-77: Zhang et al. (2018), which discussed the EASM precipitation changes in the monsoonal China during the weakening AMOC, may be cited here.

Line 129: The sentence "Stalagmite YX262 was deposited under the condition of isotope equilibrium" should be moved to the end of line 133 as a conclusion.

For the figure 1, what does the background color in the map indicate? If it's meaningful, please add a legend.

References:

Zhang, H., Brahim, Y. A., Li, H., Zhao, J., Kathayat, G., Tian, Y., Baker, J., Wang, J., Zhang, F., and Ning, Y.: The Asian Summer Monsoon: Teleconnections and Forcing Mechanisms-A Review from Chinese Speleothem $\delta$18O Records, Quaternary, 2, 26, 2019.

Zhang, H., Cheng, H., Cai, Y., Spötl, C., Kathayat, G., Sinha, A., Edwards, R. L., and Tan, L.: Hydroclimatic variations in southeastern China during the 4.2 ka event reflected by stalagmite records, Climate of the Past, 14, 1805-1817, 2018.

He, S., Y. Gao, F. Li, H. Wang, and Y. He (2017), Impact of Arctic Oscillation on the East Asian climate: A review, Earth-Science Reviews, 164, 48-62, doi: https://doi.org/10.1016/j.earscirev.2016.10.014.

Zhao J Y, Cheng H, Yang Y, Tan L, Spötl C, Ning Y, Zhang H, Cheng X, Sun Z, Li X, Li H, Liu W, Edwards R L. 2018. Reconstructing the western boundary variability of the Western Pacific Subtropical High over the past 200 years via Chinese cave oxygen isotope records. Clim Dyn, 52: 3741–3757.

---

## Author Comment (AC1) · 4 Nov 2019

The comment was uploaded in the form of a supplement:
https://www.clim-past-discuss.net/cp-2019-93/cp-2019-93-AC1-supplement.zip

---

## Author Comment (AC2) · 4 Nov 2019

The comment was uploaded in the form of a supplement:
https://www.clim-past-discuss.net/cp-2019-93/cp-2019-93-AC2-supplement.zip

---

## Author Response (AR1)

**Reviewer #1**

Dear Editor Dr. Yin and Anonymous Reviewers:

We really appreciate your time and efforts that you have spent in reading, reviewing and handling our manuscript. Your comments and suggestions have greatly improved our manuscript. Following these insightful comments and suggestions, we have conducted a point-to-point revision as listed below. We have reproduced the reviewers' comments in blue fonts, and our responses in black fonts directly below the comments. We hope that our revised manuscript is now considered to be suitable for publication with your high standard journal.

**Main Comments:**

1. The speleothem oxygen-isotope has high resolution, thus, its upward or downward trends over some specific time periods need to be quantified using the trend test methods e.g. Mann-Kendall non-parametric trend test. Moreover, the magnitude and amplitude of the EASM intensity also need to be calculated.

Reply: Many thanks for your suggestion. The upward trend during the MCA has been constrained through a linear fit method in the stalagmite $\delta^{18}O$ record and other climatic reconstructions (see new Fig. 4) in the revised manuscript. The magnitude and amplitude of the stalagmite $\delta^{18}O$ records have not been calculated. This is because the strength of MCA, LIA and CWP is not globally coherent, and in addition, our Yongxing record does not span the whole MCA and CWP periods. Most importantly, stalagmite $\delta^{18}O$ records in each cave do not have the same absolute values over the same period and their fluctuations are not coherent as well because of the different rainfall mixing extent in the epikarst zone.

2. According to the background of the co-authors, a more mechanism of EASM variation should be discussed. e.g. how the NAO effects the East Asian summer monsoon, based on some instrumental datasets or CMIP5 datasets.

Reply: Thanks a lot for pointing out this. More mechanisms of EASM variation have been added based on some instrumental datasets. We have added the following text in our revised manuscript:
"An analysis of instrumental data indicates that the winter NAO signal can be transmitted to East Asia through a wave train bridge and leads to a drier southern China but slightly wetter central China (Sung et al., 2006). On the other hand, Wu et al. (2009) have proposed that NAO-related spring SST anomalies in the North Atlantic can produce anomalous anticyclonic circulations over the Okhotsk Sea, which help to enhance the subtropical monsoon front."

3. The other proxy reconstruction from tree-ring [Liu et al., 2019] and historical documentary [Ge et al., 2008] are suggested to cross check the speleothem EASM reconstruction. Moreover, a detailed and independent local temperature reconstruction should be used to explore the relationship between the speleothem oxygen-isotope and the temperature, e.g. [Cook et al., 2013; Shi et al., 2015; Zhang et al., 2018].

Reply: The other proxy reconstructions from tree-ring (Liu et al., 2019) and historical document (Ge et al., 2009) have been utilized to cross check the speleothem $\delta^{18}O$ record. Moreover, local temperature reconstructions have been utilized to make a comparison with our stalagmite $\delta^{18}O$ record. However, our stalagmite $\delta^{18}O$ record does not show a significant correlation with these climatic reconstructions (see Figs. 1 and 2 below in our response).

[Figure]

Fig. 1 Comparisons of the Yongxing δ$^{18}$O record with other proxy reconstructions. (a) Meiyu rain from historical documents (Ge et al., 2008); (b) Yongxing δ$^{18}$O record (this study); (c) Precipitation reconstruction from tree-ring (Liu et al., 2019).

[Figure]

Fig. 2 Comparison of (c) the Yongxing $\delta^{18}$O record and temperature reconstructions by (a) Zhang et al., 2018 and (b) Shi et al., 2015.

**Specific Comments:**

1. Page 1, Lines 25-26. What meaning is the 'EASM intensity'? the amplitude or magnitude of the EASM variation?

Reply: The EASM intensity means the magnitude of the EASM itself. Based on a recent study of Zhang et al., (2018), we have rephrased the "EASM intensity" to Meiyu rain in our revised manuscript.

2. Page 2, Line 49. It is 'Mann'. Moreover, Chen (2018) is not the temperature reconstruction.

Reply: Many thanks for your corrections, we have changed 'Man' to 'Mann' and deleted 'Chen (2018)' in our revised manuscript.

3. Page 2, Lines 63-67. The detained review about the disagreement of the influence ENSO on EASM is very interesting and suggested to help the following analysis.

Reply: Many thanks for your positive comments here.

4. Page 3, Lines 78-79. The 'direct evidence' is not rigorous, even it is still difficult to connect the AMOC and EASM for the instrumental dataset analysis.

Reply: Thank you for your comments. We have revised the original expression as "available empirical data is still rare to explore the potential link between the AMOC and regional precipitation during the MCA and LIA intervals" in our revised manuscript.

5. Page 4, Lines 132-133. There is a large discrepancy between YX262 and YX275 in the early 1600s. A discussion of this difference is suggested to indicate a stable condition of the isotope.

Reply: Thank for your suggestion. We have added some discussion as follows: "A minor difference exists between the two stalagmite $\delta^{18}O$ records. The YX262 record shows a larger shift toward more negative values than the YX275 record in the early 1600s. Different feeding systems for both the stalagmites probably produce the $\delta^{18}O$ discrepancy. Longer mixing of meteorological rain within the overlying bedrock may dampen the overall rain $\delta^{18}O$ amplitude and therefore lead to the calcite $\delta^{18}O$ offsets (Tan et al., 2019; Carolin et al., 2013). Overall, the good replication between the two records suggests that the YX262 $\delta^{18}O$ signal is less influenced by the kinetic fractionation and is primarily of climatic origin." in our revised manuscript.

6. Page 5, Lines 141-142. The EASM intensity is not equal to the local rainfall, e.g. the increasing meiyu rainfall means the weak EASM.

Reply: Thank you for your suggestion. We have revised this inaccurate expression. 'the EASM intensity' has been revised as 'Meiyu rain'.

7. Page 6. Lines 188-189. The statement is inaccurate, since the north-drought and south-flood can be affected by the same factor from the instrumental analysis.

Reply: Thanks for your comment. We suggest that the north drought and south flood could result from meridional migration of the Meiyu rain belt (Yu and Zhou., 2007; Zhou et al., 2009; Zhang et al., 2018). We have added this statement in our revised manuscript.

8. Page 7, Lines 214-215. In fact, the EASM becomes weak since the late 1970s [Wang, 2001].

Reply: The EASM shows remarkable multi-decadal variability during the 20th century. The weakening of the EASM since the late 1970s attracts wide attention (Wang., 2001; Zhou et al., 2009). Recently, it is suggested that the EASM has been recovering since the early 1990s (Liu et al., 2012). Here our stalagmite record shows that the EASM increase step by step since the end of LIA on the centennial scale, generally in agreement with the increasing tendency of the global temperature. We interpret the weakening of the EASM between 1970s-1990s as a portion of the EASM multi-decadal variability, which punctuated the centennial EASM increasing since the LIA.

9. Pages 7-8. Lines 229-235. When you check Walker cell, the position of ascending or sinking branch is also important for atmosphere transport.

Reply: Thank you for your suggestion. We have considered your comments here in our revised manuscript.

10. Pages 9-10, Lines 278-311. The relationship between NAO and EASM during the CWP is complex, why it is stable during the LIA or MCA. Is a possible reason the uncertainty of NAO reconstruction?

Reply: The correlation seems better between the NAO and EASM over the MCA and LIA than the

CWP. For example, a maximum monsoon rainfall centered at 1900AD corresponds to a more negative NAO index, contradicting to the relationship between them over the MCA and LIA. The proxy-based NAO index (Trouet et al., 2009) used in the context can be consolidated by the instrumental NAO index series (Jones et al., 1997). The varied relationship over the CWP may depend on timescales as well. A better relationship occurs on centennial scales, rather than on decadal or shorter timescales.

**References**

Carolin S. A. et al., Varied response of western Pacific hydrology to climate forcings over the last glacial period. Science, 340, 1564–1566, 2013.

Cobb, K. M., Charles, C. D., Cheng, H. & Edwards, R. L. El Nino/Southern Oscillation and tropical Pacific climate during the last millennium, Nature, 424, 271–276, 2003.

Ge, Q., Guo, X., Zheng, J., Hao, Z.: Meiyu in the middle and lower reaches of the Yangtze River since 1736, Chinese Sci. Bull., 53, 107-114, 2008.

Graham, N. E. et al. Tropical Pacific—mid-latitude teleconnections in medieval times. Clim. Change, 83, 241–285, 2007.

Jones, P., Jonsson, T., Wheeler, D.: Extension to the North Atlantic oscillation using early instrumental pressure observations from Gibraltar and south-west Iceland, Int. J. Climatol., 17, 1433-1450, 1997.

Liu, H., Zhou, T., Zhu, Y., Lin, Y.: The strengthening East Asian summer monsoon since the early 1990s, Chinese Sci. Bull., 57, 1553-1558, 2012.

Liu, Y., Cai, W., Sun, C., Song, H., Cobb, K., Li, J., Leavitt, S., Wu, L., Cai, Q., Liu, R., Ng, B., Cherubini, P., Büentgen, U., Song, Y., Wang, G., Lei, Y., Yan, L., Li, Q., Ma, Y., Fang, C., Sun, J., Li, X., Chen, D., Linderholm, H.: Anthropogenic aerosols cause recent pronounced weakening of Asian Summer Monsoon relative to last four centuries, Geophys. Res. Lett., 46, 5469-5479, 2019.

Sung, M., Kwon, W., Baek, H., Boo, K., Lim, G., Kug, J.: A possible impact of the North Atlantic Oscillation on the east Asian summer monsoon precipitation, Geophys. Res. Lett., 33, L21713, 2006.

Tan, L., Shen, C-C., Löwemark, L et al.: Rainfall variations in central Indo-Pacific over the past 2700y, P Natl Acad Sci USA, 17201-17206, 2019, 10.1073/pnas.1903167116.

Trouet, V., Esper, J., Graham, N., Baker, A., Scourse, J., and Frank, D.: Persistent Positive North Atlantic Oscillation Mode Dominated the Medieval Climate Anomaly, Science, 324, 78-80, 2009.

Wang, H.: The weakening of the Asian Monsoon Circulation after the End of 1970's, Adv. Atmos. Sci., 18, 376-386, 2001.

Wu, Z., Wang, B., Li, J., Jin, F.: An empirical seasonal prediction model of the east Asian summer monsoon using ENSO and NAO, J. Geophys. Res., 114, D181120, 2009.

Yan, H., Sun, L., Wang, Y., Huang, W., Qiu, S., and Yang, C.: A record of the Southern Oscillation Index for the past 2,000 years from precipitation proxies, Nat. Geosci., 4, 611-614, 2011

Yu, R., Zhou, T.: Seasonality and Three-Dimensional Structure of Interdecadal Change in the East Asian Monsoon, J. Climate, 20, 5344-5355, 2007.

Zhang, H., Griffiths, M., Chiang, J., Kong, W., Wu, S., Atwood, A., Huang, J., Cheng, H., Ning, Y.,

Xie, S.: East Asian hydroclimate modulated by the position of the westerlies during Termination I, Science, 362, 580-583, 2018.

Zhou, T., Gong, D., Li, J., Li, B.: Detecting and understanding the multi-decadal variability of the East Asian Summer Monsoon-Recent progress and state of affairs, Meteorol. Z., 13, 455-467, 2009.

**Reviewer #2**

Dear Editor Dr. Yin and Anonymous Reviewers:

We really appreciate your time and efforts that you have spent in reading, reviewing and handling our manuscript. Your comments and suggestions have greatly improved our manuscript. Following these insightful comments and suggestions, we have conducted a point-to-point revision as listed below. We have reproduced the reviewers' comments in blue fonts, and our responses in black fonts directly below the comments. We hope that our revised manuscript is now considered to be suitable for publication with your high standard journal.

**General comments:**

(1) In the introduction, in order to better introduce research background to readers, it is necessary to add some more references. For example, after the sentence of Lines 46-48, Lines 49-51 and Lines 61-63. In addition, some references are not properly used (not the most proper one).

Reply: According to your suggestion, we have added more references in lines 46-48, lines 49-51 and lines 61-63 in the revised manuscript, and deleted some references.

(2) Please add more information about the stalagmite sample. For instance, the sample image can be added in the figure 2, including subsample locations of U-Th dates and if possible stable isotopes as well. Was the whole sample (YX262) or only one portion analyzed in this study? What is the mineral of the sample? The authors only mentioned 'calcite record' in the discussion (Line 241).

Reply: Thank you for your suggestions. We have added a sample image in Figure 2 and marked the sampling locations for U-Th dates in the image in our revised manuscript. The whole sample (YX262) is analyzed in our study, and it is composed of white opaque to brown transparent calcite.

(3) For discussion 4.1, in Lines 153-156, "Thus, the stalagmite d18O signal reflects the regional summer monsoon intensity. . .. . .", how to understand the term "regional summer monsoon intensity"? In addition, the authors should always point out the timescale when they discuss the significance of the stalagmite d18O proxy. In Lines 146-148, "two most recent studies have reconciled these two contradictory interpretations. . .. ..", it sounds like that the two studies already resolved the debates of the Chinese stalagmite d18O proxy. There are many papers that have addressed to some extent this issue recently, such as Zhao et al., 2018 and Zhang et al., 2019. Additionally, it appears that the cave d18O was considered to be the 'monsoon intensity' and local precipitation as well at different places, lacking a consistency.

Reply: Many thanks for your detailed comments. Recently published review articles greatly enlighten our understanding on how to interpret the stalagmite $\delta^{18}O$ records in East Asia (Zhang et al., 2019; Cheng et al., 2019). We have re-organization the related content in the revised manuscript and added those new references.

(4) The small amplitude changes in stalagmite d18O value may have complicated mechanisms behind, such as temperature effect, amount effect, source changes, upper stream rainout, and evaporations etc. If explained solely as local rainfall amount, please provide a comparison to the instrumental record for each cave record or cite related published papers.

Reply: Based on your suggestion, we have made a comparison between our stalagmite $\delta^{18}O$ record and Meiyu reconstruction from historical documents (Ge et al., 2008). The stalagmite $\delta^{18}O$ record matches the Meiyu rain well on decadal to centennial timescales. When the stalagmite $\delta^{18}O$ is lighter, the Meiyu rain weakens, and vice versa. We interpret our Yongxing $\delta^{18}O$ record as the indicator of the Meiyu rain variation.

(5) In the section 4.2, the authors should compare their record with the stalagmite record from Heshang Cave, which is fairly close to Yongxing Cave. I suggest adding the Heshang d18O record in the figure 3, and have a related discussion in the section. In addition, I strongly encourage the authors to compare the Yongxing record with local historical records or cite related papers, which may provide a validation test on the interpretation of the Yongxing d18O record.

Reply: Many thanks for your suggestion. We have added the Heshang $\delta^{18}O$ record in Figure 4 of our revised manuscript and included a related discussion about their variations therein. In addition, we have compared our $\delta^{18}O$ record with instrumental precipitation and Meiyu rain reconstruction. Our record shows a good correlation with the Meiyu reconstruction on decadal to centennial timescales (Ge et al., 2008). No significant correlation is found between the Yongxing $\delta^{18}O$ record and instrumental precipitation as well as temperature records at the Yichang station (see Fig. 1 below in our response). This relationship was also inferred in a comparison with the Heshang record (He et al., 2009).

[Figure]

Fig. 1 Comparison of the Yongxing $\delta^{18}O$ time-series and other records. The black line represents the

Yongxing δ¹⁸O time-series; the magenta line indicates Meiyu rain reconstructed from historical documents (Ge et al., 2008); the red and blue lines indicate the instrumental precipitation and temperature at the nearby Yichang station, respectively.

(6) When comparing the d18O values between different time periods (the MCA, LIA and CWP), the differences of the mean values should be provided. The trends of records, as well as similarities between different records are merely visually defined. Statistical methods should be considered to show their significances.

Reply: Many thanks for your suggestion. The mean δ¹⁸O values between the MCA, LIA and CWP represent mean hydrological conditions over these episodes. Our Yongxing δ¹⁸O record does not cover the whole MCA and CWP episodes. Moreover, the MCA and LIA are not globally coherent (Neukom et al., 2019). Thus, onsets and terminations of these episodes are difficult to unambiguously defined. In our study, we compare δ¹⁸O minima (drier condition) during the MCA and CWP periods, two warm intervals. The δ¹⁸O minima could add valuable information to assessing the natural and anthropogenic forcing in central China. The trends of our records have been constrained through the linear fit methods in the revised manuscript. In addition, correlation analyses have been utilized to show significances between two records.

(7) In sections 4.4 and 4.5, I suggest that the authors analyze the relationship of the local precipitation (and/or d18O) at Yongxing Cave site with ENSO, NAO, PDO and AMOC indexes (reconstructed from instrumental data).

Reply: Following your suggestion, we have made a comparison of the Yongxing δ¹⁸O record with ENSO, NAO and PDO indexes reconstructed from instrumental data (see Fig. 2 below in our response). No significant correlation is found between them. The AMOC index reconstructed from instrumental data began to exist since 2004, with no temporal overlap with our record. {Since 2004, there has been a major British-American observation project, called RAPID (http://www.rapid.ac.uk/rapidmoc/overview.php), which tries to measure the total flow at a particularly suitable latitude (26.5° North) with 226 moored measuring instruments (Meinen et al., 2019).}

[Figure]

[Figure]

[Figure]

Fig. 2 Comparisons of the Yongxing δ¹⁸O record and other instrumental data. (a), (b) and (c) show the comparisons of our Yongxing record with SOI, NAO and PDO indexes, respectively. The SOI data is from http://www.bom.gov.au/climate/current/soihtm1.shtml; the NAO index data is from https://crudata.uea.ac.uk/cru/data/nao/index.htm; the PDO index data (Jones et al., 1997) is from https://www.ncdc.noaa.gov/teleconnections/pdo/.

(8) Overall, the causal links of the stalagmite d18O records with the AMOC, NAO, ENSO as suggested by the authors are rather tentative. For example, a visual similarity between two records cannot be used to definitively validate their causal linkage.

Reply: Thanks for your suggestion. Based on the Yongxing δ¹⁸O record, we discuss a potential influence of the Pacific and North Atlantic Oceans on the Meiyu rain. The causal linkage would be examined by future geological records and climate simulations. Regrettably, climate simulation is beyond the scope of our current study. Nevertheless, correlation analyses have been calculated to validate causal linkages in the revised manuscript.

**Specific comments:**

1. Lines 24-26: I don't think we can say the "EASM intensity is similar in both northern and central China, . . .. . .". I mean we cannot say local EASM intensity instead of local precipitation amount. In addition, the timescale should be always mentioned.

Reply: Thanks very much for your guidance. Following the guidance, we have revised the inaccurate description in lines 24-26 in the reorganized manuscript.

2. Lines 31 and 278: The authors use "surprisingly" twice in the manuscript. Actually, many studies already found the North Atlantic climate can influence the EASM changes, for example He et al. (2017).

Reply: Thanks for your suggestion. The word of "surprisingly" is inaccurate in lines 31 and 278. Therefore, we have deleted the word in our revised manuscript.

3. Lines 74-77: Zhang et al. (2018), which discussed the EASM precipitation changes in the monsoonal China during the weakening AMOC, may be cited here.

Reply: Thanks for pointing out this important research. We have referred to Zhang et al., (2018) in our revised manuscript.

4. Line 129: The sentence "Stalagmite YX262 was deposited under the condition of isotope equilibrium" should be moved to the end of line 133 as a conclusion.

Reply: Many thanks. The sentence has been moved to the end of line 133 as a conclusion.

5. For the figure 1, what does the background color in the map indicate? If it's meaningful, please add a legend.

Reply: Thanks for your advice, we have modified the figure and added a legend.

**References**

Cheng, H., Zhang, H., Zhao, J., Li, H., Ning, Y., Kathayat, G.: Chinese stalagmite paleoclimate researches: A review and perspective, Sci. China Earth Sci., 62, 1489-1513, 2019.

Ge, Q., Guo, X., Zheng, J., Hao, Z.: Meiyu in the middle and lower reaches of the Yangtze River since 1736, Chinese Sci. Bull., 53, 107-114, 2008.

He, L., Hu, C., Huang, J., Xie, S., Wang, Y.: Characteristics of large-scale circulation of East Asian Monsoon indicated by oxygen isotope of stalagmite, Quaternary Science, 29, 950-956, 2009 (in Chinese with English abstract).

Jones, P., Jonsson, T., Wheeler, D.: Extension to the North Atlantic oscillation using early instrumental pressure observations from Gibraltar and south-west Iceland, Int. J. Climatol., 17, 1433-1450, 1997.

Meinen, C., Johns, W., Moat, B., Smith, R. Johns, E., Rayner, D., Frajka-Williams, E., Garcia, R., Garzoli, S.: Structure and Variability of the Antilles Current at 26.5°N, Journal of Geophyscial Research, https://doi.org/10.1029/2018JC014836, 2019.

Neukom, R., Steiger, N., Gómez-Navarro, J., Wang, J., Werner, J.: No evidence for globally coherent warm and cold periods over the preindustrial Common Era, Nature, 571, 550-554, 2019.

Zhang, H., Brahim, Y., Li, H., Zhao, J., Kathayat, G., Tian, Y., Baker, J., Wang, J., Zhang, F., Ning, Y., Edwards, R., Cheng, H.: The Asian Summer Monsoon: Teleconnections and Forcing Mechanisms-A Review from Chinese Speleothem δ18O Records, Quaternary, 2, 26, 2019.

Zhang, H., Cheng, H., Cai, Y., Spötl, C., Kathayat, G., Sinha, A., Edwards, R. L., and Tan, L.: Hydroclimatic variations in southeastern China during the 4.2 ka event reflected by stalagmite records, Climate of the Past, 14, 1805-1817, 2018.

**Hydrological variations in central China over the past millennium**

**and their links to the Tropic Pacific and North Atlantic Oceans**

Fucai Duan[a,*], Zhenqiu Zhang[b,c,*], Yi Wang[d,e,*], Jianshun Chen[a], Zebo Liao[c], Shitao

Chen[c], Qingfeng Shao[c], Kan Zhao[c]

[a]College of Geography and Environmental Sciences, Zhejiang Normal University,
Jinhua 321004, China
[b]School of Life Sciences, Nanjing Normal University, Nanjing 210023, China
[c]College of Geography Science, Nanjing Normal University, Nanjing 210023, China
[d]Department of Geography and School of Global Studies, University of Sussex, Falmer,
Brighton BN1 9QJ, UK
[e]Department of Earth System Science, Institute for Global Change Studies, Tsinghua
University, Beijing 100084, China

*Corresponding authors:
E-mail addresses: fcduan@foxmail.com (F. Duan), zhangzhenqiu163@163.com (Z.
Zhang), yi.wang@sussex.ac.uk (Y. Wang)

**Abstract:** Variations of precipitation, aka the Meiyu rain, in East Asian summer
monsoon (EASM) domain during the last millennium could help enlighten the
hydrological response to future global warming. Here we present a precisely dated and
highly resolved stalagmite $\delta^{18}O$ record from the Yongxing Cave, central China. Our
new record, combined with a previously published one from the same cave, indicates
that the Meiyu rain has changed dramatically in association with the global temperature
change. In particular, our record shows that the Meiyu rain has weakened during the
Medieval Climate Anomaly (MCA) and the Current Warm Period (CWP), but
intensified during the Little Ice Age (LIA). We find that the Meiyu rain is similarly
wetter during the MCA and CWP in northern China and similarly drier in central China,
but relatively wetter during the CWP in southern China. This discrepancy indicates a
complicated localized response of the regional precipitation to the anthropogenic
forcing. The weakened (intensified) Meiyu rain during the MCA (LIA) matches well
with the warm (cold) phases of Northern Hemisphere surface air temperature. This
Meiyu rain pattern also corresponds well with the climatic conditions over the Tropical
Indo-Pacific warm pool. On the other hand, our record shows a strong association with
the North Atlantic climate as well. The reduced (increased) Meiyu rain correlates well
with positive (negative) phases of North Atlantic Oscillation. In addition, our record
links well with the strong (weak) Atlantic meridional overturning circulation during the
MCA (LIA) period. All above-mentioned localized correspondences and remote
teleconnections on decadal to centennial timescales indicate that the Meiyu rain is
coupled closely with oceanic processes in the Tropical Pacific and North Atlantic
Oceans during the MCA and LIA.

**Keywords:** Stalagmite; East Asian summer monsoon; Global warming; Last

Millennium; Little Ice Age; Medieval Climate Anomaly; the Meiyu rain

**1    Introduction**

The last millennium was climatically characterized by the Medieval Climate

Anomaly (MCA; 900-1400 AD) and the Little Ice Age (LIA; 1400-1850 AD), and the

Current Warm Period (CWP; 1850AD to present). These three episodes attract broad attention within the scientific and policy-making communities, because they contain critical information to distinguish between the natural and anthropogenic climate variability. Origins of the MCA and LIA are attributed to the radiative forcing associated with solar activities and volcanic eruptions, yet the CWP is considered as a result of increasing anthropogenic greenhouse gases (Bradley and Jonest.,1993; Hegerl et al., 2007; Lamoureus et al., 2001; Sigl et al., 2014). In particular, the CWP is much warmer than the MCA (PAGES 2k Consortium, 2013). In association with the global temperature change, East Asian summer monsoon (EASM) precipitation has changed significantly (Paulsen et al., 2003; Zhang et al., 2008; Tan et al., 2009, 2011a, 2015).

Many studies have indicated that monsoonal climate of China has generally recorded wetter MCA and drier LIA in the north, but show reverse conditions in the south (Tan et al., 2009, 2018; Chen et al., 2015). However, it is unclear about the hydrological variation during the MCA and LIA over central China. Moreover, less is known about the relative intensity of precipitation between the CWP and MCA, two recent warm periods. The examination of the relative precipitation intensity is the key to evaluating the hydrological responses under the anthropogenic warming.

To better understand hydrological responses to the anthropogenic warming, it is necessary to appreciate the natural forcing of the hydrological cycle during the MCA

and LIA periods before the greenhouse gas emission. The hydroclimate in the EASM

domain is strongly influenced by the Tropical Pacific and North Atlantic Oceans (Wang et al., 2005; Zhang et al., 2018a, Cheung et al., 2018). The Tropical Pacific Ocean feeds the warm and moisture air directly into the EASM domain, and therefore exerts a strong influence (Karami et al., 2015). Several studies have indicated that the hydrological condition in the EASM domain is affected by alternations of La Nina-like and El Nino- like conditions in the Tropical Pacific during the last millennium (e.g., Chen et al., 2015;

Zhao et al., 2016; Zhang et al., 2018a). However, these studies did not reach an agreement on how the Tropical Pacific affects hydrological change in the EASM

domain. To precisely understand a spatio-temporal evolution of the hydrological cycle,
we need to know exactly which changes in the hydrological cycle are linked to which
modes of the Pacific atmosphere-ocean circulation during the MCA and LIA in central
China. On the other hand, the North Atlantic signal can be transmitted to other parts of
the world through the Atlantic meridional overturning circulation (AMOC; Bond et al.,
2001). Marine sedimentary records have suggested that strong (weak) AMOC over the
warm Greenland interstadials (stadials) correlated tightly with intervals of enhanced
(reduced) EASM during the last glaciation (Wang et al., 2001a; Jiang et al., 2016).
Similarly, weak EASM episodes occurred in association with ice-rafted events in the
North Atlantic, which is capable of weakening the AMOC during the Holocene (Wang
et al., 2005; Zhao et al., 2016; Zhang et al., 2018b). This covariation implies a persistent
influence of the AMOC on EASM. However, available empirical data is still rare to
explore the potential link between the AMOC and regional precipitation (e.g., EASM)
during the MCA and LIA intervals.

Here we present a new precisely-dated and highly-resolved stalagmite record from
Yongxing Cave, Central China. This record, together with a recently published records
from the same cave (Zhang W et al., 2019), advances our understanding of the
hydrological cycle in East Asia during the last millennium.

**90  2   Materials and methods**

Two stalagmites (YX262 and YX275) are used in this study, both from Yongxing
Cave (31°35′N, 111°14′E; elevation 800 m above msl; Fig. 1), central China. The
previously published stalagmites YX275 has reported detailed variability in the EASM
since the LIA (Zhang W et al., 2019). The new candle-like stalagmite YX262 is 159
mm long and 55 mm wide. It is composed of white opaque to brown transparent calcite
(see Fig. 2). The Yongxing Cave is located between the Chinese Loess Plateau and the
Yangtze River. Average annual rainfall is about 1000 mm at the site of the cave.
Atmospheric temperature is about 14.3°C and relative humidity is close to 100% inside
the cave. The cave site is climatically influenced by East Asian Monsoon, featured with
wet and warm summer, and dry and cold winter.

Stalagmite YX262 was first halved and then polished for the purpose of the
subsequent sampling. For stable isotope analyses, powdered subsamples, weighing
about 50-100 µg, were drilled on the polished surface along the central growth axis of the stalagmite. A total of 159 subsamples were obtained at 1 mm increments. The $\delta^{18}O$

measurements were performed on a Finnigan-MAT-253 mass spectrometer at Nanjing

Normal University. Results are reported as per mil (‰) against the standard Vienna Pee

Dee Belemnite (VPDB). Precision of $\delta^{18}O$ is 0.06‰ at the 1-sigma level. For U-Th dates, six powdered subsamples, about 100 mg each, were drilled along the central growth layer. Procedures for chemical separation and purification of uranium and thorium were described in Shao et al. (2017). U and Th isotope measurements were performed on a Neptune MC-ICP-MS at Nanjing Normal University. All the dates are in stratigraphic order with uncertainty of less than 3% of the actual dates (see Table 1).

**3 Results**

**3.1 Chronology**

The six U-Th dates and corresponding isotopic rations are shown in Table 1.

Adequate uranium concentrations (0.5–0.7 ppm) and low initial thorium contents (200–

700 ppt, with the exception of 1440 ppt) produced precise dates with small age uncertainty (6–20 years). The chronology for the stalagmite was established by the

StalAge algorithm (Scholz and Hoffmann, 2011). The age model shows that the stalagmite YX262 was deposited from 1027 to 1639 AD (see Fig. 2). The age-depth plot indicates the growth rate of the stalagmite is stable, reaching 0.26 mm/year. The high and stable growth rate suggests that the stalagmite grew continuously without a significant hiatus. Visual inspections consolidate the continuity of the stalagmite growth.

The temporal resolution is 3.8 year, allowing for detailed characterizing the Asian hydroclimate for the first half of the second millennium.

**3.2 Stable isotope**

The $\delta^{18}O$ record of YX262 displays a pronounced fluctuation during the whole period (see Fig. 4). The $\delta^{18}O$ values ranges from -9.31‰ to -7.88‰, averaging -8.60‰.

The $\delta^{18}O$ values decrease gradually from 1027 to 1372 AD, and then increase gradually before rapidly increasing to the $^{18}O$-enriched conditions from 1515 AD. The interval with high $\delta^{18}O$ values is ~100-year long, which is terminated by a pulse to more negative values at 1626 AD. In general, the $^{18}O$-depleted interval is coeval with the

MCA and the $^{18}O$-enriched interval corresponds to the early LIA (see Fig. 4).

**4 Discussion**

**4.1 The interpretation of our $\delta^{18}O$**

Stalagmite YX262 was deposited under the condition of isotope equilibrium. Relative to the Hendy tests, replication tests have been considered as a more vigorous method to examine the isotope equilibrium (Dorale and Liu, 2009). The YX262 $\delta^{18}O$ record matches another Yongxing cave record during the overlapping interval (see Fig. 5; Zhang W et al., 2019), indicating an equilibrium condition for the isotope. A minor difference exists between the two stalagmite $\delta^{18}O$ records. The YX262 record shows a larger shift toward more negative values than the YX275 record in the early 1600s. Different feeding systems for both the stalagmites probably produce the $\delta^{18}O$ discrepancy. Longer mixing of meteorological rain within the overlying bedrock may dampen the overall rain $\delta^{18}O$ amplitude and therefore lead to the calcite $\delta^{18}O$ offsets (Tan et al., 2019; Carolin et al., 2013). The more negative $\delta^{18}O$ shift occurred at the beginning of the growth of stalagmite YX262. At the beginning, the mixing of meteorological rain within the overlying bedrock is low, resulting to the lighter $\delta^{18}O$ values. Overall, the good replication between the two records suggests that the YX262 $\delta^{18}O$ signal is less influenced by the kinetic fractionation and is primarily of climatic origin. Nevertheless, the climatic significance of the cave $\delta^{18}O$ record in eastern China remains a long-term scientific debate. For example, the $\delta^{18}O$ records were considered to reflect changes in moisture sources (so-called "circulation effect", Tan., 2014, 2016), moisture pathways (Baker et al., 2015), and a combination of the EASM and winter temperature (Clemens et al., 2010, 2018). Two recent review articles have greatly enlightened our understanding of the stalagmite $\delta^{18}O$ records in the EASM domain (Zhang H et al., 2019; Cheng et al., 2019). They have proposed that the cave $\delta^{18}O$ records have reflected large-scale and integrated changes in the Asian summer monsoon intensity on the orbital and millennial scales. This interpretation is supported by strong correlations among the cave $\delta^{18}O$ records across China (e.g., Yuan et al., 2004; Zhao et al., 2010; Cheng et al., 2009; 2016) and correlations with climate conditions in major global climate systems, such as Antarctica, Greenland and Westerly climate (see Figs. 2 and 3 in Cheng et al., 2019). The lighter stalagmite $\delta^{18}O$ values signify larger rainout along the moisture trajectory and thus stronger EASM intensity and vice versa. However, the interpretation of the stalagmite $\delta^{18}O$ records remains complex on the annual to centennial scales, due to a wide range of potential influencing factors, such as summer rainfall, moisture sources and seasonality of precipitation (Zhang H et al.,

2019; Cheng et al., 2019). At the Dongge cave location, stalagmite $\delta^{18}O$ records were interpreted to be associated with the monsoon precipitation on the decadal to centennial scale, because of its covariation with the local hydrological proxy, annual band thickness (Zhao et al, 2015). Our Yongxing $\delta^{18}O$ record in central China correlates well with Meiyu rain fluctuations in the middle and lower reaches of the Yangtze River on the decadal to centennial scale (see Fig. 3; Ge et al., 2008). When the stalagmite $\delta^{18}O$

values are lighter, the Meiyu rain is lower and vice versa. This relationship is further supported by inverse correlations of stalagmite $\delta^{18}O$ records with local rainfall variation (trace element ratio and $\delta^{13}C$) in the nearby Haozhu Cave (Zhang et al., 2018c). As suggested in Zhang et al. (2018c) and Cheng et al. (2019), increased (weakened) EASM

would lead to a shorter (longer) Meiyu rain stage and thus a decrease (increase) of precipitation in the middle and lower reaches of the Yangtze River. Thus, the Yongxing

$\delta^{18}O$ signal mainly reflects Meiyu rain conditions on the decadal to centennial scales, with lower and higher $\delta^{18}O$ values reflecting decreased and increased rainfall, respectively.

**4.2 The regional characters of the MCA and LIA**

The climate condition during the MCA and LIA has been extensively studied for the monsoonal China (e.g., Chen et al., 2015; Xu et al., 2016; Tan et al., 2018). In general, wetter in the north and drier in the south were inferred during the MCA and the opposite during the LIA (e.g., Chen et al., 2015; Tan et al., 2018). The boundary between the north and south of China was estimated to be about along the River Huai at 34°N (Chen et al., 2015), the modern geographical dividing line between northern and southern China. As an interesting exception, the Dongge cave records in Guizhou,

Southwestern China (25°17′N, 108°5′E) showed a wetter MCA and drier LIA (see Fig.

4; Wang et al., 2005; Zhao et al., 2015). This is consistent with strong spatiotemporal variability of precipitation in the broad EASM region.

Our Yongxing record, slightly south to 34°N, is further supported by the nearby

Heshang $\delta^{18}O$ record, despite larger chronological offsets between them (see Fig. 4; Hu et al., 2008). Both stalagmite $\delta^{18}O$ records consistently show a trend toward lighter values over the MCA period and a double valley structure over the LIA period. An extra comparison shows that the Yongxing and Heshang (Hu et al., 2008) $\delta^{18}O$ records in central China vary broadly in phase with the Dongge record in the south, as well as

Wanxiang (Zhang et al., 2008) and Huangye (Tan et al., 2011b) records in the north. These cave records indicate a drier MCA and wetter LIA in central China, but the opposite in the north and south (see Fig. 4). Again, a minor but important discrepancy exists between these cave records during the MCA. The cave records in the south display an increasing precipitation trend, but those in the norther and central China reflect a decreasing trend during the MCA (see Fig. 4 for the trends indicated by the arrows). To explain this discrepancy, we compare all the cave records to changes in temperatures of Northern Hemisphere (Mann et al., 2009) and northern China (Tan et al., 2003), and meridional displacement of the Intertropical Convergence Zone (ITCZ; Haug et al., 2001). The result indicates that all the cave records collectively exhibit a broad similarity to the variation in the temperatures and the displacement of the ITCZ (see Fig. 4). Detailed inspection displays that the weakening precipitation signal recorded in the northern caves during the MCA is linked with the decreasing temperatures in the Northern Hemisphere and northern China. In contrast, the intensifying signal recorded in the southern cave during the MCA corresponds to the northward displacement of the ITCZ. The comparison indicates that the different climate patterns between the south and north may result from different controlling factors at lower and higher latitudes, respectively. The 'north drought' and 'south flood' can result from meridional migration of the Meiyu rain belt (Yu and Zhou., 2007; Zhou et al., 2009; Zhang et al., 2018c). It seems that the cold temperature from the north restrains the northward migration of the Meiyu rain belt related to the movement of the ITCZ during the MCA, leading to the hydrological seesaw between the northern and central China. It is noted that the enhanced precipitation condition documented in the Dongge records is contradictory with those reported in many other paleoclimate records in the south. For example, drier MCA and wetter LIA were suggested in an integrated stalagmite $\delta^{18}O$ record from Sichuan Province (Tan et al., 2018), a pollen-derived rainfall record near the Yongxing Cave site (He et al., 2003), and a lake-based rainfall record in Guangdong Province (Chu et al., 2002). This regional discrepancy can be checked by additional highly-resolved and precisely dated records in southern China.

**4.3   The hydrological condition during the MCA as compared to the CWP**

A comparison of the relative intensity of precipitation between the MCA and CWP could be useful to evaluate the hydrological response towards the current global warming. Many studies have found that the CWP is much warmer than the MCA on global and hemispheric scales (Bradley et al., 2003; Mann et al., 2008, 2009; PAGES 2k Consortium, 2013). With regard to the hydrological response, northern China shows an increased or comparable precipitation maximum during the MCA as compared to the CWP (e.g., the Wanxiang and Huangye Caves' records in Fig. 4). A similar precipitation minimum is documented in the Yongxing and Heshang records in central China (see Fig. 4). However, two Dongge records in southern China collectively shows a slight decrease in precipitation maximum during the MCA as compared to the CWP (see Figs. 4, 5). This is indicated by a 0.39‰ higher $\delta^{18}O$ maximum during the MCA than the CWP (see Fig. 5). The increased precipitation during the CWP relative to the MCA is parallel to the global temperature evolution, in particular in the western Pacific Warm Pool region (Chen et al., 2018). This correspondence supports the hypothesis that current global warming intensifies the Asian summer monsoon (Wang et al., 2013). The intensified Asian summer monsoon was suggested due to strong coupling of the climate system related to the global warming. Wang et al. (2013) have stated a mega ENSO condition could trigger a stronger EASM in the CWP through the intensified Hadley and Walker circulations. On the other hand, southern China is partially influenced by the Indian Ocean, which also brings moisture to the area of our study (An et al., 2011). We suggest the small discrepancy between Yongxing and Dongge records could be due to the different localized effects in southern China as Dongge Cave is much closer to Indian Ocean than Yongxing Cave.

[revised manuscript text omitted]

**4.5   The link to the North Atlantic Climate**

Our Yongxing record shows a good correlation with the North Atlantic climate. As illustrated in Fig. 7, the decreased (increased) Meiyu rain during the MCA (LIA) coincides with a persistent positive (neutral to slightly negative) North Atlantic Oscillation index (NAO; Trouet et al., 2009; see Fig. 7c; R=-0.19; P<0.05; N=182). In addition, these Meiyu rain variations resemble changes of the Atlantic meridional overturning circulation (AMOC), measured by the drift ice index (see Fig. 7d; Bond et al., 2001) and mean grain size of sortable silt (see Fig. 7e; Thornalley et al., 2018; R=0.39; P<0.01; N=186) in the North Atlantic. The decreased Meiyu rain corresponds to the strong AMOC during the MCA and the increased Meiyu rain to the weak AMOC during the LIA. These strong correlations indicate an influence of the NAO and AMOC on the EASM. During the MCA, positive NAO induces a warmer winter in Europe, which reduces snow accumulation over Eurasia and therefore allows for a farther penetration inland of the EASM next summer (Overpeck et al., 1996). An analysis of instrumental data indicates that the winter NAO signal can be transmitted to East Asia through a wave train bridge and leads to a drier southern China but slightly wetter central China (Sung et al., 2006). On the other hand, Wu et al. (2009) have proposed that NAO-related spring SST anomalies in the North Atlantic can produce anomalous anticyclonic circulations over the Okhotsk Sea, which help to enhance the subtropical monsoon front. Robust AMOC can intensify the EASM through northward positioning the ITCZ (Wang et al., 2017). During the LIA, weaker NAO and AMOC would produce decreased EASM in the reversed fashion. It has been proposed that conditions of the NAO are dynamically coupled to states of the AMOC (Trouet et al., 2009; Wanamaker et al., 2012). The strong (weak) NAO during the MCA (LIA) contributes to enhanced (weakened) AMOC through enhancing (weakening) the westerly (Trouet et al., 2009). On the other hand, solar activity is usually considered as the root trigger of natural climate change. The Yongxing record is broadly similar to changes in solar irradiance (Steinhilber et al., 2009; see Fig. 7a). The decreased Meiyu rain is paralleled with the greater solar activity during the MCA and the increased Meiyu rain with the less solar activity during the LIA. The solar forcing of the Meiyu rain variation, dependent on the EASM strength (Zhang et al., 2018c), can be conducted through modulating the Asia-Pacific temperature contrast (Kutzbach et al., 2008), the AMOC intensity (Wang et al., 2005) and the ENSO condition (e.g., Asmerom et al., 2007; Zhao et al., 2016). However, relative importance of these forcing pathways is unknown and, most importantly, the ENSO condition remains a matter of debate during the last millennium (e.g., Cobb et al., 2003; Yan et al., 2011a). As a counterpart to the MCA, the CWP is similarly marked by decreased Meiyu rain, strong AMOC and high solar output (see Fig. 7). However, the relationship between the Meiyu rain and NAO condition is not significant during the CWP, with the decreased Meiyu rain failing to match the expected more positive NAO. Longer term data from instrumental observations and historical proxies is needed to assess the linkage between NAO condition and Meiyu rain during the CWP.

**5 Conclusions**

Based on a new and a recently published stalagmite records from the Yongxing cave, central China, we reconstruct a continuous evolutional history of the Meiyu rain during the past millennium and link its variation with the Pacific and North Atlantic climates. The climatic characters in our record are generally antiphase with those in the Wanxiang and Huangye cave records in northern China. The decreased (increased) Meiyu rain during the MCA (LIA) correlates with the warm (cold) surface temperature and enhanced (reduced) rainfall over the IPWP. Based on the strong correlation with the ENSO reconstruction, our records support an El Nino-like condition during the MCA and a La Nina-like condition during the LIA. In addition, our records show a potential link between the Meiyu rain and the North Atlantic climate. The decreased Meiyu rain coincides with substantially positive NAO and robust AMOC during the MCA, while the increased Meiyu rain corresponds with neutral to negative NAO and weak AMOC during the LIA.

**Data availability**

Data in this study are available on request.

**Competing interests**

The authors declare that they have no conflict of interest.

**Author contributions**

FD, ZZ and YW designed the study and wrote the manuscript. FD, ZZ, YW and JC revised the manuscript. FD, QS, ZL and KZ performed [230]Th dating and oxygen isotope measurements. FD and SC collected samples. All authors discussed the results and contributed to the manuscript.

**Acknowledgments**

This work was supported by Zhejiang Provincial Natural Science Foundation (no. LY19D020001) and National Natural Science Foundation of China grants (nos. 41602181, 41572340 and 41572151). We are grateful to Editor Dr. Qiuzhen Yin and two anonymous reviewers for their helpful suggestions and comments that have greatly improved this manuscript.

**Table and figure**

Table l U-series dating results of stalagmite YX262 from Yongxing Cave

| Sample depth (mm) | $^{238}U$ (ppb) | $^{232}Th$ (ppt) | $\delta^{234}U$ (measured) | $^{230}Th/^{238}U$ (activity) | $^{230}Th$ Age (a) (uncorrected) | $\delta^{234}U_{initial}$ (corrected) | $^{230}Th$ Age (a) (corrected) |
|---|---|---|---|---|---|---|---|
| YX262-5 | 546.0 ±0.5 | 307.9 ±0.6 | 607.5 ±1.0 | 0.006230157 ±0.00014 | 423.5 ±9.4 | 608.2 ±1.0 | 413.1 ±10.8 |
| YX262-25 | 595.5 ±0.3 | 280.5 ±0.6 | 790.6 ±1.9 | 0.00788248 ±0.00008 | 481.0 ±5.1 | 791.7 ±1.9 | 473.1 ±6.3 |
| YX262-48 | 506.3 ±0.3 | 281.6 ±0.5 | 762.1 ±1.9 | 0.009468079 ±0.00010 | 587.3 ±6.4 | 763.4 ±1.9 | 577.9 ±8.0 |
| YX262-75 | 517.7 ±0.3 | 724.3 ±0.1 | 680.5 ±2.1 | 0.010930422 ±0.00010 | 711.3 ±6.4 | 681.8 ±2.1 | 686.3 ±13.9 |
| YX262-95 | 651.8 ±0.3 | 1448.0 ±0.3 | 806.5 ±2.0 | 0.013146471 ±0.00010 | 796.0 ±6.3 | 808.3 ±2.0 | 759.4 ±19.1 |
| YX262-116 | 583.4 ±0.8 | 283.0 ±0.4 | 956.6 ±1.0 | 0.014987259 ±0.00012 | 838.0 ±6.6 | 958.9 ±1.0 | 830.8 ±7.5 |

Decay constant values are $\lambda_{234}$=2.82206×10$^{-6}$a$^{-1}$, $\lambda_{238}$=1.55125×10$^{-10}$a$^{-1}$, $\lambda_{230}$=9.1705×10$^{-16}$a$^{-1}$ and $\delta^{234}U$ =
([$^{234}U/^{238}U$]$_{activity}$-1)×1000. Corrected $^{230}Th$ age calculation, indicated in bold, is based on an assumed initial
$^{230}Th/^{232}Th$ atomic ratio of (4 ± 2) × 10$^{-6}$. All corrected dates are years before 2017 A.D.

[Figure]

Fig.1 Schematic climate setup of East Asian Monsoon and our study site. The blue star and black triangles represent Yongxing Cave in central China and other caves in the monsoonal region, respectively.

[Figure]

Fig. 2 The age versus depth model, image and $\delta^{18}O$ record for our stalagmite YX262. The small black dots and vertical error bars indicate $^{230}Th$ dates and errors of these dates, respectively. The big black dots represent the locations of $^{230}Th$ dates. The middle green line indicates the model age, and upper and lower red lines indicate the age in 95% confidence level, respectively.

[Figure]

Fig.3 A comparison between the Yongxing δ¹⁸O record (blue line) and reconstructed
Meiyu rain (pink line; 7-years running average; Ge et al., 2008).

[Figure]

Fig. 4 A comparison of the Yongxing δ¹⁸O time-series with other proxy records. (a)

Northern Hemisphere reconstructed temperature (Mann et al., 2009); (b) Northern China reconstructed temperature (Tan et al., 2003); (c) Huangye Cave δ18O composite (Tan et al., 2011); (d) Wanxiang Cave δ18O record (Zhang et al., 2008); (e) Heshang Cave δ18O record (Hu et al., 2008); (f) Yongxing Cave record (this study); (g) Dongge Cave record (Wang et al., 2005; Zhao et al., 2015); (h) Cariaco Basin Ti content record (Haug et al., 2001). Light yellow and blue bars indicate the MCA and LIA, respectively. Arrows, constrained by linear fit methods, indicate trends of the climatic variations.

[Figure]

Fig. 5 The relative intensity of Meiyu rain during the MCA as compared to the CWP. The upper panel is the Dongge cave record (blue curve, Wang et al., 2005); the lower panel is the Yongxing Cave YX262 (red curve) and YX275 (green curve, Zhang W et al., 2019) records. On average, the Dongge Cave record shows a 0.39‰ lower δ18O value during the CWP than the MCA. However, the Yongxing record shows a comparable value between the CWP and MCA.

[Figure]

Fig. 6 A comparison between Meiyu rain and Pacific climate. (a) Yongxing cave record (this study); (b)Tropical Pacific rainfall record (Oppo et al., 2009); (c) Tropical Pacific $\delta^{18}O$ record (Oppo et al., 2009); (d) Tropical Pacific sea surface temperature (Oppo et al., 2009); (e) Red colour intensity in southern Ecuador (Moy et al., 2002); (f) Hydrological reconstruction of ENSO from Tropical Pacific (Yan et al., 2011a). Yellow, grey and orange bands represent the MCA, LIA, and CWP, respectively.

[Figure]

Fig. 7 A comparison among Meiyu rain, solar activity and North Atlantic climate. (a) Total solar irradiance (Steinhilber et al., 2009); (b) Yongxing Cave record (this study); (c) North Atlantic Oscillation index (Trouet et al., 2009); (d) North Atlantic drift ice index (Bond et al., 2001); (e) Sortable silt grain size in the North Atlantic (Thornalley et al., 2018). Yellow, grey and orange bands represent the MCA, LIA, and CWP, respectively.

---

## Author Response (AR2)

Dear Editor Dr. Yin and Anonymous Reviewers:

We really appreciate your time and efforts that you have spent in reading, reviewing and handling our manuscript. Your comments and suggestions have greatly improved our manuscript. Following these insightful comments and suggestions, we have conducted a point-to-point revision as listed below. We have reproduced the reviewers' comments in blue fonts, and our responses in black fonts directly below the comments. We hope that our revised manuscript is now considered to be suitable for publication with your high standard journal.

**Main comments:**

1. It should be a district heavy phase during the Medieval Climate Anomaly in the speleothem YX262 δ18O record, which includes a maximum peak around AD 1050. This indicates a strong Meiyu rain. How to obtain a weak Meiyu rain in Medieval Climate Anomaly (Page 1, lines 22-23)? The minimum peak around AD 1400 should be re-determined from an age control point.

Reply: Many thanks for your comment.

The YX262 $\delta^{18}O$ record shows a gradually decreasing trend over the MCA (~800-1400 AD, see Fig. 3). The decreasing $\delta^{18}O$ record indicates a weakening Meiyu rain state. In the revised manuscript, we have rephrased the original inaccurate expression in line 22-23. The rephrased sentence is " In particular, our record shows that the Meiyu rain has been weakened during the Medieval Climate Anomaly (MCA), but intensified during the Little Ice Age (LIA). During the Current Warm Period (CWP), our record indicates a similar weakening of the Meiyu rain."

The minimum peak around AD 1400 is indeed well constrained by two $^{230}$Th dates. It is constrained by dates 1330.7 ± 13.9 year at 75 mm and 1439.1 ± 8.0 year at 48 mm (please see Table 1 in the manuscript).

2. A detailed explanation is necessary for the relationship between δ18O record and East Asian summer monsoon through the Meiyu rain, and the sentence (Page 5, lines 163-164) is unclear.

Reply: Thanks for your suggestion. The relationship was well described in a recently published paper (Zhang et al., 2018, Science), referred to in our manuscript. Here we reproduce the description to show a detailed explanation.

"The seasonal rainfall evolution over East Asia undergoes a number of quasi-stationary intraseasonal stages with abrupt transitions in between. During spring, persistent rainfall in southern China is followed by substantial convection over the South China Sea during the pre-Meiyu stage in mid-May. By mid-June, the Meiyu begins and rainfall shifts to central China, and around mid-July, the rain band shifts again to be located over northeast China, marking the onset of the midsummer stage, which terminates around mid-August."

"Rainfall changes over East Asia arise through changes in the transition timing and duration of the EASM intraseasonal stages." Increased (weakened) EASM, further penetrating inland, would lead to a shorter (longer) Meiyu rain stage and thus a decrease (increase) of precipitation in the middle and lower reaches of the Yangtze River. On the other hand, the EASM intensity is broadly measured by the stalagmite $\delta^{18}O$ variation, which reflects changes in the fraction of water vapor rained out between tropical oceans and cave locations (Zhang H et al., 2019; Cheng et al., 2019). Thus, the stalagmite $\delta^{18}O$ variation is associated with the changes in the EASM and Meiyu rain, with lower and higher $\delta^{18}O$ stalagmite values reflecting decreased and increased Meiyu rain, respectively.

We rephrased the unclear sentence at line 163-164, Page 5. The rephrased sentence is "The stalagmite $\delta^{18}O$ values reflect changes in the fraction of water vapor precipitated out between tropical oceans and cave locations (Zhang H et al., 2019; Cheng et al., 2019). A strengthened (weakened) EASM indicates increased (decreased) rainout along the moisture trajectory, and therefore lighter (heavier) stalagmite $\delta^{18}O$ values (Zhang H et al., 2019; Cheng et al., 2019)."

3. I am not a native English, but English needs to be carefully improved, e.g. 'and vice versa' is not often used in English.

Reply: Many thanks for your instruction. We have carefully polished English, and removed "and vice versa".

**Specific Comments:**

1. An age control point is usually found on the bottom of the published speleothem records, which is also important to determine the age of the minimum peak of the YX262 record during the Medieval Climate Anomaly.

Reply: Many thanks for your suggestion. As aforementioned in the response to Main Comment 1, the minimum peak of the YX262 record around AD 1400 is well anchored by two $^{230}$Th dates, namely, 1330.7 ± 13.9 year at 75 mm and 1439.1 ± 8.0 year at 48 mm. Personally, the expected bottom date is not needed in this study. Conversely, the bottom age is indispensable at the time of dating the onset of the stalagmite growth.

2. The link between the Meiyu rain and Northern Hemisphere temperature is unclear to me.

Reply: As described in Zhang et al. 2018, the "jet transition hypothesis" (Chiang et al., 2015) links the Meiyu rain with Northern Hemisphere temperature. The hypothesis proposes that rainfall changes over East Asia arise through changes in the transition timing and duration of the EASM intraseasonal stages. When the Northern Hemisphere temperature decreases, the temperature gradient between the East Asia and tropical oceans declines, resulting in an earlier northward positioning of the westerlies relative to the Tibetan Plateau. The earlier northward would lead to an earlier termination of the Meiyu stage and prolonged midsummer stage, and thus a weakened Meiyu and strengthened EASM (Zhang et al., 2018).

3. A north-south diploe mode of the speleothem δ18O records during the Medieval Climate Anomaly period is a new information to me, which is consistent with the results from the historical documents. I suggest that the authors double check the records, since others studies shows a monopole mode in the speleothem δ18O records [Tan, 2016].

Reply: Many thanks for your suggestion. We clarify the paradox in the following words. The study of (Tan, 2016) shows a monopole climate mode during the Medieval Climate Anomaly (MCA) based on several stalagmite records from various areas of China. These stalagmite $\delta^{18}O$ records collectively display a lighter value over the MCA. Our YX262 $\delta^{18}O$ record also shows such a lighter value. Here our detailed study finds different long-term trends between the northern and southern stalagmite $\delta^{18}O$ records within the MCA period (see Fig. 3 in the manuscript). This new finding is not in contradiction with the conclusion of (Tan, 2016).

4. It is suggested to check the description of results for current warm period. The reason is that the current warm period includes the instrumental period. e.g., the authors considered that the speleothem record in Yongxing cave can represent the intensity of East Asian summer monsoon, and it means that there is an extremely increase in the YX275 record during the early of 19th century, which need evidence from the early instrumental data or other high-resolution proxy data. To the best of my knowledge, there is no significant evidence of such a strong East Asian summer monsoon based on the instrumental records.

Reply: Thanks for your suggestion. The Current Warm Period (CWP) begins from the end of the Little Ice Age to present. The CWP is usually defined as the interval from 1850 AD to present (Mann et al., 2009). As such, the CWP does not overlap the early 19th century. In addition, we could not recognize an extremely EASM increase from the YX275 record during the early 19th century. In contrast, a multidecadal-scale weak monsoon event is recognized over the period of 1800-1850AD in the YX275 record. Indeed, an extremely EASM increase occurred over the early 20th century in the YX275 record, although the EASM began to decrease since the late 20th century. This extremely EASM increase is further supported by precisely-dated Wanxiang Cave record (Zhang et al., 2008) and the annually-laminated Dongge Cave record (Zhao et al., 2015).

5. Page 2, lines 42-44. The reference is suggested to define these periods.
Reply: Many thanks for your suggestion. References (Lamb., 2002; Mann et al., 2009) have been added.

6. Page 4, line 128. There is no Fig. 3 before Fig. 4.
Reply: Many thanks for pointing out the mistake. We have exchanged the two figures in order.

7. In Fig.3, a direct comparison of the raw data with 3.8 year temporal resolution is interesting to show.
Reply: Many thanks for your comment. The 3.8-year-resolution YX262 $\delta^{18}$O record is raw data in this figure, which is not smoothed.

8. A sententious physical process is suggested to explain the link between the Meiyu and the Indo-Pacific warm pool, the Meiyu and the Northern Atlantic Oscillation or Atlantic meridional overturning circulation.
Reply: Thank you for your suggestion. Sententious physical processes have been inserted in Section 4.4 and 4.5 to clearly link the Meiyu to the Indo-Pacific warm pool and the North Atlantic climates. The added sentences are "A northward migration or expansion of the ITCZ over the Indo-Pacific warm pool would strengthen the EASM and shorten the Meiyu stage. Conversely, a southward migration or contraction of the ITCZ would weaken the EASM and prolong the Meiyu stage (Zhang et al., 2018)." and "A strong EASM, resulting from the strong NAO and AMOC, would shorten a Meiyu rain stage. Conversely, a weak EASM, resulting from the weak NAO and AMOC, would lengthen a Meiyu rain stage (Zhang et al., 2018)."

**References**

Zhang, H., Griffiths, M., Chiang, J., Kong, W., Wu, S., Atwood, A., Huang, J., Cheng, H., Ning, Y., Xie, S.: East Asian hydroclimate modulated by the position of the westerlies during Termination I, Science, 362, 580-583, 2018.

Zhang, H., Brahim, Y., Li, H., Zhao, J., Kathayat, G., Tian, Y., Baker, J., Wang, J., Zhang, F., Ning, Y., Edwards, R., Cheng, H.: The Asian Summer Monsoon: Teleconnections and Forcing Mechanisms-A Review from Chinese Speleothem $\delta^{18}O$ Records, Quaternary, 2, 26, 2019.

Cheng, H., Zhang, H., Zhao, J., Li, H., Ning, Y., Kathayat, G.: Chinese stalagmite paleoclimate researches: A review and perspective, Sci. China Earth Sci., 62, 1489-1513, 2019.

Chiang, J., Fung, I., Wu, C., Cai, Y., Edman, J., Liu, Y., Day, J., Bhattacharya, T., Mondal, Y., Labrousse, C.: Role of seasonal transitions and westerly jets in East Asian paleoclimate, Quaternary Science Reviews, 108, 111-129, 2015.

Tan, M.: Circulation background of climate patterns in the past millennium: Uncertainty analysis and re-reconstruction of ENSO-like state, Science China Earth Sciences, 59(6), 1225-1241, 2016.

Mann, M., Zhang, Z., Rutherford, S., and Bradley, R., Hughes, M., Shindell, D., Ammann, C., Faluvegi, G., and Ni, F.: Global Signatures and Dynamical Origins of the Little Ice Age and Medieval Climate Anomaly, Science, 326, 1256-1260, 2009.

Zhang, P., Cheng, H., Edwards, R., Chen, F., Wang, Y., Yang, X., Liu, J., Tan, M., Wang, X., Liu, J., An, C., Dai, Z., Zhou, J., Zhang, D., Jia, J., Jin, L., and Johnson, K.: A test of climate, sun, and culture relationships from an 1810-year Chinese cave record, Science, 322, 940-942, 2008.

Zhao, K., Wang, Y., Edwards, R., Cheng, H., Liu, D., and Kong, X.: A high-resolved record of the Asian Summer Monsoon from Dongge Cave, China for the past 1200 years, Quaternary Sci. Rev., 122, 250-257, 2015.

Lamb, H.: Climate, History and the Modern World, London, Routledge, https://doi.org/10.4324/9780203433652, 2002.